# Histopathological change of age-related hearing loss in female advance-aged CBA/CaJ mice

Yoshiaki Inuzuka[1], Kunio Mizutari[1,2]*, Takaomi Kurioka[1,3], Jun Suzuki[4], Yutaka Koizumi[5], Koji Araki[1], Akihiro Shiotani[1]

1 Department of Otolaryngology, Head and Neck Surgery, National Defense Medical College, Tokorozawa, Saitama, Japan, 2 Department of Otolaryngology-Head and Neck Surgery, Tokyo Women's Medical University, Shinjuku, Tokyo, Japan, 3 Department of Otorhinolaryngology and Head and Neck Surgery, Kitasato University School of Medicine, Sagamihara, Kanagawa, Japan, 4 Department of Otolaryngology, Head and Neck Surgery, Tohoku University Graduate School of Medicine, Sendai, Miyagi, Japan, 5 Department of Otolaryngology, Head and Neck Surgery, Yamagata University Faculty of Medicine, Yamagata, Yamagata, Japan

* tari@mbf.ocn.ne.jp

## Abstract

With the increase in the older population, the number of individuals with age-related hearing loss is also growing explosively. Therefore, there is an urgent need to identify the detailed pathology of age-related hearing loss and develop novel treatment strategies. In this study, we have investigated the audiological physiology and cochlear pathology of advanced-age CBA/CaJ mice, a strain that resists early pathological hearing loss. The subjects were naturally aged close to their lifespan limit (> two years) under normal in vivo conditions. We used 11 CBA/CaJ mice aged between 129 and 138 weeks to establish an aged group. To compare the electrophysiological function and histological changes, a young group was established using 12 young mice aged between 9 and 14 weeks. The loss of outer hair cells peaked at 11.3 kHz, and the greatest synapse loss was observed in the 5.6 kHz region, which was covered by the dominant frequency in the ambient sound. Furthermore, atrophy and microthrombus formation occurred in the stria vascularis, with endolymphatic hydrops observed in the cochlear apical turn. In the spiral ganglion and cochlear nerve, a reduction in the number of cells was accompanied by morphological changes indicative of cell aging. Increased levels of derivative-reactive oxygen metabolites, an oxidative stress marker, were observed in aged mice. These results indicate that age-related hearing loss involves a combined pathology of acoustic cochlear damage, which is potentially associated with chronic sound exposure and metabolic changes owing to mitochondrial dysfunction and oxidative stress accumulation. Accordingly, these two distinct etiologies must be addressed to prevent and treat age-related hearing loss.

**Data availability statement:** All relevant data are within the paper and its Supporting information files.

**Funding:** This work was supported by a grant for Advanced Defense Medical Research provided by the Japanese Ministry of Defense (Grant Number A-4, A.S.) and two grants from JSPS KAKENHI (Grant Numbers 21K09573, K.M. and 20K18263, T.K.) The funders had no role in study design, data collection and analysis, decision to publish, or preparation of the manuscript.

**Competing interests:** The authors have declared that no competing interests exist.

## Introduction

Globally, the aging population is growing progressively. According to the World Health Organization (WHO) fact sheets, it is estimated that 2 billion individuals worldwide will be over the age of 60 years by 2050, and more than 25% of them will experience age-related hearing loss [1].

Age-related hearing loss not only causes an increase in hearing thresholds and a decrease in speech discrimination, especially in noisy environments [2,3], but also reduces the quality of life owing to tinnitus [4] and is a risk factor for dementia and depression [5,6]. Therefore, it is crucial to elucidate the detailed pathology of age-related hearing loss and develop novel treatment strategies.

Several studies have investigated age-related changes in the cochlea. Schuknecht has reported the relationship between age-related pathological classification of the cochlea (sensory, neural, strial, and cochlear conductive) and audiograms by examining human temporal bone pathological specimens [7,8]. Subsequently, research on the pathophysiology of age-related hearing loss has focused on lesions at various sites, including a decrease in endocochlear potential (EP) and associated dysfunction in hair cells (HCs), a reduction in synapses and axons between inner hair cells (IHCs) and cochlear nerves [9–12], mitochondrial DNA mutations and associated cellular metabolic dysfunction, and accumulation of oxidative stress [13].

Previous animal studies have frequently employed genetically modified early-aging mice and early hearing loss mouse strains (especially C57BL/6 mice). In the CBA/CaJ mouse strain, the age-related hearing threshold increases progressively, which is similar to that observed in humans, making it suitable for evaluating age-related changes in the cochlea during its lifespan [11,14]. Additionally, several studies on age-related hearing loss have focused on individual cochlear components, such as HCs, synapses, stria vascularis (SV), and spiral ganglion neurons (SGNs). However, few studies have analyzed these components comprehensively; in particular, reports on advanced-aged mice at the end of their lifespan are lacking. To develop appropriate treatment and preventive interventions for hearing loss in older adults, it is crucial to perform a detailed pathophysiological analysis of advanced-age animals. Therefore, in the current study, we performed a comprehensive physiological and histological evaluation of age-related changes in hearing and cochleae in advanced-aged CBA/CaJ mice, which had aged naturally in a vivarium.

## Materials and methods

### Animals

Herein, we used 11 female CBA/CaJ mice aged between 129 and 138 weeks (29–31 months), which were deemed the aged group. CBA/CaJ mice were purchased from Jackson Laboratory (Bar Harbor, ME, USA) and bred in the animal faculty at Tohoku University. To compare electrophysiological functions and histological changes, a young group of 12 female CBA/CaJ mice aged 9–14 weeks was established. The animals were provided access to a regular diet and water *ad libitum*. Mice in the aged group were raised under an identical acoustic environment for the duration of their

lifespans, i.e., under 76.7 dB sound pressure level (SPL) as an equivalent continuous sound level with Z-frequency weighting (LZeq) and stable frequency characteristics (Fig 1). The sound level was measured at three points of rearing shelves using a precision sound level meter (NL-52A, RION, Tokyo, Japan) and calculated the average. These sound assessments were conducted multiple times during their lifespan, and the results were found to be identical. Aged and young animals were shipped to National Defense Medical College for experiment. All experimental protocols were approved by the Institutional Animal Care and Use Committee of National Defense Medical College (approved no. 21051). All efforts were made to minimize animal suffering and the number of animals used in the study.

## Cochlear function test

We used the same system to measure cochlear electrophysiological function as previously reported [15]. Cochlear function tests were conducted at six log-spaced frequencies (half-octave steps from 5.6 to 32.0 kHz) with a LabVIEW-driven data-acquisition system in an acoustically and electrically shielded chamber. Mice (n = 10 ears of 5 animals per group) were anesthetized intramuscularly using ketamine (75 mg/kg) and medetomidine (1 mg/kg). Needle electrodes were subcutaneously inserted at the vertex (reference) and pinna (measuring) to detect the auditory brainstem responses (ABRs), with a ground electrode placed at the back. ABRs were evoked using monaurally 5 ms tone pips (0.5-ms rise-fall with a cos2 onset envelope delivered at 31/s). The sound level was increased in 5 dB increments from 20 to 80 dB SPL. At each sound level, 512 responses were averaged using alternating stimulus polarities. ABR thresholds were defined as the lowest level (≥20 dB SPL) at which any wave could be detected using ABR Peak Analysis Software (https://www.masseyeandear.org/research/otolaryngology/eaton-peabody-laboratories/engineering-core) and visual inspection of the stacked waveforms by blinded reviewers for the group labeling. When no response was observed at the highest available sound level, the threshold was designated as 5 dB greater (i.e., 85 dB SPL). The software also identified the wave I peak (P1) and calculated the wave I peak-to-peak amplitude and peak latency. The amplitude and latency at 80 dB SPL were measured and analyzed to compare the reactions of the cochlear nerve under identical stimulation. If ABR thresholds were 85 dB SPL, the frequencies of each ear were excluded from the ABR P1 analysis, whereas the value of 85 dB SPL was included for threshold analysis.

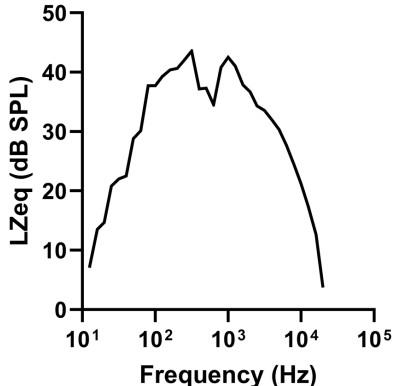

**Environmental noise**

**Fig 1. The frequency-analyzed sound level of the environmental ambient sound surrounding the mice.** The graph shows the distribution of the sound level by frequency. The peak is bimodal at 315 and 1000 Hz, inclined to the low frequency in the audible range of mice. The average LZeq, LZFmax, and LZFmin at the rearing shelf of mice are 76.7, 82.0, and 71.5 dB SPL, respectively. LZeq, Equivalent continuous sound level with Z-frequency weighting; LZFmax, maximum time-weighted sound level measured with Z-frequency weighting and fast time weighting; LZFmin, minimum time-weighted sound level measured with Z-frequency weighting and fast time weighting; SPL, sound pressure level.

To measure the distortion product otoacoustic emissions (DPOAEs) at 2f1-f2, primary tones were set such that the frequency ratio (f2/f1) was 1.2 and the f2 level was 10 dB below the f1 level. For each f2/f1 primary pair, the levels were swept in increments of 5 dB from 20 dB to 80 dB SPL (for f2). Waveform and spectral averaging were used at each level to increase the signal-to-noise ratio of the recorded ear canal sound pressure, and the DPOAE amplitude at 2f1-f2 was extracted from the averaged spectra along with the noise floor at the adjacent points of the spectrum. The threshold was defined as the f2 level required to produce a DPOAE at 0 dB SPL.

## Cochlear immunohistochemistry

The immunostaining protocol has been described previously [15]. Mice (n = 3 ears of 2 animals in each group) were euthanized with an overdose of ketamine (100–150 mg/kg) and medetomidine (2–3 mg/kg) intramuscularly and perfused with phosphate-buffered saline (PBS), followed by 4% paraformaldehyde (PFA) transcardially. For cochlear whole mount dissection, the cochlea was post-fixed for 2h in PFA 4% at 4°C and decalcified in 0.5 M EDTA (pH 7.5) overnight at 4°C. The cochleae were dissected into four pieces and incubated in 1% Triton X-100 and 5% normal horse serum at room temperature for 1h. The specimens were incubated overnight at 37°C with the following primary antibodies: rabbit anti-Myo7a polyclonal antibody at 1:200 (#25–6790, Proteus Biosciences, Waltham, MA, USA), purified mouse IgG1 anti-CtBP2 monoclonal antibody at 1:200 (#612044, BD Transduction Labs, San Jose, CA, USA), and mouse IgG2a anti-GluR2 monoclonal antibody at 1:2000 (#MAB397, Millipore, Burlington, MA, USA). The specimens were washed thrice with PBS, followed by the application of secondary antibodies and incubation twice for 1h at 37°C. Alexa Fluor 350-conjugated goat anti-rabbit IgG (1:200) (#A21068; Life Technologies, Carlsbad, CA, USA), Alexa Fluor 488-conjugated goat anti-mouse IgG2 (1:500) (#A21131; Life Technologies), and Alexa Fluor 568-conjugated goat anti-mouse IgG1 (1:1000) (#A21124; Life Technologies) were used. Primary and secondary antibodies were diluted in 1% Triton X-100 and 5% normal horse serum, respectively.

The specimens were washed thrice with PBS and mounted on slides containing an antifade medium (VECTASHIELD; Vector Laboratories, Burlingame, CA, USA). To construct a cochlear frequency map with a quick review of the whole mounting and observation of cochlear HCs, specimens were examined under a fluorescence microscope (BZ-X700, Keyence Corporation, Osaka, Japan), while specimens were examined under a fluorescence microscope (TCS SP8, Leica, Wetzlar, Germany) to observe cochlear synapses. Cochlear lengths were measured for each specimen, and a cochlear frequency map was computed to precisely localize the IHCs in the 5.6, 8.0, 11.3, 16.0, 22.6, and 32.0 kHz regions, respectively. ImageJ software (NIH, Bethesda, MD, USA) and the ImageJ plugin (https://masseyeandear.org/research/otolaryngology/eaton-peabody-laboratories/histology-core) were used to measure the total length of the cochlear whole mounts and individual segments.

## Cochlear plastic sections and transmission electron microscopy (TEM)

Specimens of cochlear cross sections and TEM were prepared as previously reported [16]. The mice (n = 8 ears of 4 animals per group) were anesthetized and transcardially perfused with PBS, followed by 2% PFA and 2% glutaraldehyde in 0.1 M PB (pH = 7.4). The cochleae were harvested and bathed overnight in the same fixative at 4°C. After decalcification in 0.5 M EDTA pH 7.5 overnight at 4°C, cochleae were then osmicated (1% OsO4), dehydrated in ethanol and propylene oxide, embedded in Epon and hardened in a 60°C oven for two days.

Specimens were sliced along a horizontal plane parallel to the cochlear modiolus. For quantitative assessment of SGNs, auditory nerve fibers (ANFs; i.e., peripheral axons of cochlear nerve), SV, and the area of scala media (SM), the tissue was sectioned at the osseous spiral lamina (OSL) near the habenula perforate and at a near-mid-modiolar plane with a glass knife (1μm) using a Leica Ultracut R microtome, stained with toluidine blue, and mounted on slides. Specimens were viewed under a light microscope (BZ-X700). The same surfaces were sectioned for TEM scanning using an Ultracut N microtome (Reichert-Nissei) at 80 nm thicknesses and collected onto mesh grids. Each grid cell was

counterstained using uranyl acetate and lead citrate. Samples were viewed using a transmission electron microscope (JEM-1400Plus; JEOL, Tokyo, Japan) at an accelerating voltage of 100 kV. TEM images of SGNs, ANFs, and SV were used for quantitative and morphological assessments.

## Histopathological analysis

HC and synapse analyses were performed as described previously [16,17]. We calculated the percentage of surviving IHCs and outer HCs (OHCs). The numbers of surviving HCs per 200 μm length (X-Y plane) at each frequency were counted. The survival rate of HCs in the aged group was calculated by normalization to that of the young group.

For cochlear synaptic assessment, confocal high-resolution z-stack images of the IHC area stained with CtBP2, GluR2, and Myo7a were obtained using an oil immersion objective lens (63x) with a 3.1 x digital zoom and a 0.25 μm step size under confocal fluorescence microscopy, as aforementioned. The image stacks were imported to ImageJ, and the CtBP2/GluR2 puncta per IHC were counted within a 50 μm range (X-Y plane) in each image stack. The number of synapses was manually counted by visualizing the colocalization of CtBP2 and GluR2. Orphan ribbons were counted as CtBP2 puncta without adjacent GluR2 immunoreactivity.

Morphometric assessments of the SV and SM were performed, as described previously [18,19]. The SV, cochlear turn, SM, and scala vestibuli areas at the same near-mid-modiolar plane were quantified using ImageJ for the basal, middle, and apical cochlear turns, respectively. The estimated cochlear frequency regions of those were 6–8 kHz at apical, 12–16 kHz at middle, and 24–32 kHz at basal turn, as referred to in the previous report [20]. In the analysis of the area of SV and SM, the subsequent quantitative metrics were used to mitigate the variability attributed to differences in the cutting orientation among the cochlear specimens, referring to the preceding literature [18,19].

SV area = the area of SV/ the area of cochlear turn

SM area = the area of SM/ (the area of SM + the area of scala vestibuli)

The quantitative assessment of SGNs and ANFs was performed as described previously [16]. Regarding SGN density, SGNs that exhibited a clear nucleus and cytoplasm in Rosenthal's canal of the basal, middle, and apical turns were counted. The area of Rosenthal's canal in the same plane was measured using ImageJ software. The density of SGNs per 10,000 $\mu m^2$ was calculated for each profile. To assess the SGN cell size, soma areas were measured using ImageJ in the same sections used for enumerating SGNs. Twenty SGNs were randomly selected from each section to measure and calculate the average cell size. Axonal counts were performed using OSL at both the basal and apical turns to evaluate ANF density. The estimated cochlear frequency regions of basal and apical turns for the ANF assessment were 20–64 kHz and 4–12 kHz, respectively [20]. Neuronal profiles in the fascicles of each section were determined. The number of ANFs per fascicle was divided by the fascicle area by perineurium, as measured using ImageJ. The density of ANFs was averaged from 10 OSL openings in each ear.

TEM was used to evaluate myelination, the number of mitochondria in SGNs and ANFs, myelin thickness, and axon caliber measured using ImageJ software. Data from 100 myelinated axons from each ear were evaluated to assess myelin thickness, axon caliber, and g-ratio (estimated by dividing the axon diameter by the myelinated fiber diameter). The number of myelin lamellae was evaluated from the TEM micrographs of seven myelinated axons. To assess the number of mitochondria, 10 SGNs (basal, middle, and apical turns in each ear) and 100 ANFs (basal and apical OSL openings in each ear) were randomly selected. The mitochondria were manually counted, and the average number per SGN or ANF was calculated.

## Systemic oxidative stress measurement

We measured derivative-reactive oxygen metabolites (d-ROMs) in the serum to assess systemic oxidative stress in mice. This procedure was performed as previously described [21]. In brief, whole blood samples were collected transcardially from aged and young mice under general anesthesia (n = 5 per group). Blood samples were centrifuged at 3,000 rpm for 15 min at 4°C immediately after blood sampling (< 2 minutes), followed by the supernatant serum was collected. d-ROM

values were measured using a free radical analyzer (FREE Carpe Diem, WISMERLL, Tokyo, Japan). The results are expressed as arbitrary units called Carratelli units (CARRs).

## Statistical analyses

We performed a two-way analysis of variance (ANOVA) with frequency and age as independent variables, followed by Šidák's *post hoc* test to analyze statistical differences in DPOAE and ABR measurements, HCs and synaptic counts, SV area, SM area, SGNs, and ANFs measurements. For the comparative analysis of HC and synaptic counts, SV area, SM area, SGN density, SGN cell size, SGN mitochondrial count, ANF axon density, axon caliber, axon mitochondrial number, myelin thickness, myelin lamella number, and g-ratio, the raw values were normalized by the young group, dividing the individual values by the average of the young group. In the cochlear functional test and cross-section/TEM histological analysis, two-way ANOVAs and post hoc tests were performed on both per-ear (i.e., including individual values from each ear) and per-mouse (i.e., including averaged values if data were collected from both ears per mouse) bases, respectively. A simple linear regression analysis was conducted in order to examine the correlation between the number of IHC, OHC, synapses, or orphan ribbons, and ABR P1 amplitude or latency in the young and aged groups. If a simple linear regression identified a significant correlation at any frequency, we proceeded to a multiple linear regression that included these metrics to evaluate their independent contributions. The Mann-Whitney *U* test was performed for d-ROM analysis. Effect size was calculated using Cohen's d values. All data are presented as means ± standard errors. Statistical significance was set at *p*-value <0.05. Statistical sample size calculations were performed with an α error probability of 0.05 and a power (1-β error probability) of 0.8. We referred to a previous study of the same strain to anticipate metrics of young and aged mice [11]. The difference in survival percentages of IHC, OHC, SGN, and synapse at 5.6 kHz (15%, 75%, 30%, and 35%, respectively) from two time points (16 weeks vs. 128 weeks) provided the required number for each group was less than three. Data analyses were performed using GraphPad Prism (version 9.0; GraphPad Software Inc., Boston, MA, USA).

## Results

### Assessment of cochlear function

Ten ears from both groups were evaluated for electrophysiological auditory function. DPOAE thresholds presenting 0 dB SPL in the aged group were significantly increased compared to the young group (two-way ANOVA, interaction: $F_{(5, 107)}$ = 12.29, *p* < 0.0001; Šidák *post hoc* test: *p* < 0.0001, Cohen d = 2.72 at 11.3 kHz, *p* < 0.0001, Cohen d = 5.47 at 16.0 kHz, and *p* < 0.0001, Cohen d = 3.27 at 22.6 kHz, *p* < 0.0001, Cohen d = 1.31 at 32.0 kHz; Fig 2A).

The aged group exhibited a significant ABR threshold elevation at all tested frequencies (two-way ANOVA, interaction: $F_{(5, 108)}$ = 3.62, *p* = 0.0045; Šidák post hoc test; *p* < 0.0001; Fig 2B). The effect size was robust at low and middle frequencies in ABR thresholds: the scores of Cohen d were 10.18 at 5.66 kHz, 7.76 at 8.00 kHz, 9.54 at 11.33 kHz, 12.55 at 16.0 kHz, 7.15 at 22.6 kHz, 4.31 at 32.0 kHz, respectively.

Considering ABR P1, which reflects cochlear nerve activity, the amplitude in the aged group was markedly reduced at most frequencies (two-way ANOVA, interaction: $F_{(5, 95)}$ = 3.29, *p* = 0.0088; Šidák post hoc test: *p* = 0.0015, Cohen d = 2.12 at 8.00 kHz, *p* < 0.0001, Cohen d = 2.02 at 11.33 kHz, p < 0.0001, Cohen d = 2.26 at 16.00 kHz, p = 0.0003, Cohen d = 3.08 at 22.65 kHz; Fig 2C). The ABR P1 latency showed a trend toward prolongation, with a significant difference observed at certain frequencies (two-way ANOVA, interaction: $F_{(5, 95)}$ = 2.72, *p* = 0.024; Šidák post hoc test: *p* = 0.0028, Cohen d = 2.58 at 8.00 kHz, *p* = 0.0089, Cohen d = 1.58 at 11.33 kHz, *p* = 0.019, Cohen d = 1.18 at 16.00 kHz, *p* < 0.0001, Cohen d = 2.74 at 22.65 kHz, *p* < 0.0001, Cohen d = 2.46 at 32.00 kHz; Fig 2D).

The overall results from the per-mouse analysis were consistent with those from the per-ear analysis (S1 Fig and S2 Table). However, at several frequencies, ABR P1 latency did not reach statistical significance, likely due to the reduced statistical power resulting from the smaller sample size.

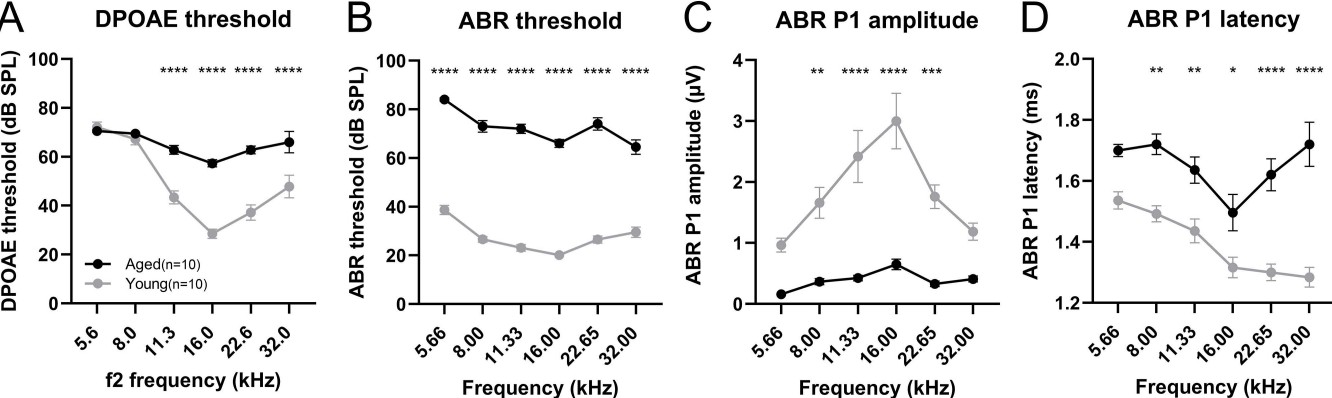

**Fig 2. Cochlear electrophysiological function results in aged and young mice.** (A) The DPOAE thresholds in the aged group are significantly elevated. (B) ABR thresholds in the aged group are significantly elevated at all frequencies. (C, D) The ABR P1, representing the summated response from the auditory nerve in the cochlea, is assessed under 80 dB SPL auditory stimulation. In the aged group, the amplitude is significantly low, and the latency is prolonged. The number of subjects is 10 ears per group. Error bars represent standard error of mean. Asterisks indicate significant differences. *$p < 0.05$, **$p < 0.01$, ***$p < 0.001$, ****$p < 0.0001$. DPOAE, distortion product otoacoustic emission; ABR, auditory brainstem response; ABR P1, ABR wave peak 1.

## Survival of cochlear HCs and synaptic ribbons

To evaluate the relevance of electrophysiological functions, we explored changes in cochlear histology (three ears in two mice from both groups). Immunofluorescence was used to assess the survival of HCs (Fig 3A). The IHCs were relatively well preserved, although a significant loss was observed at a specific frequency (two-way ANOVA, interaction: F (5, 24) = 1.56, $p = 0.21$; Šidák post hoc test: $p = 0.0081$, Cohen d = 3.00 at 16.0 kHz; Fig 3B). The OHCs exhibited a significant loss at all frequencies, with the peak OHC loss occurring at 11.3 kHz. The survival rate tended to be lower at 5.6 and 8.0 kHz than at higher frequencies (two-way ANOVA, interaction: F (5, 24) = 2.23, $p = 0.084$; Šidák post hoc test: $p < 0.0001$, Cohen d = 4.99 at 5.6 kHz, $p < 0.0001$, Cohen d = 5.47 at 8.0 kHz, $p < 0.0001$, Cohen d = 11.83 at 11.3 kHz, $p < 0.0001$, Cohen d = 5.70 at 16.0 kHz, $p = 0.0001$, Cohen d = 3.58 at 22.6 kHz, and $p = 0.0007$, Cohen d = 4.77 at 32.0 kHz; Fig 3C).

Cochlear synapses between IHCs and ANFs were evaluated using immunofluorescence (Fig 4A). Significant synaptic loss was observed at several frequencies (two-way ANOVA, interaction: F (5, 24) = 2.12, $p = 0.098$; Šidák post hoc test: $p < 0.0001$, Cohen d = 3.00 at 5.6 kHz, $p = 0.010$, Cohen d = 2.67 at 11.3 kHz, $p = 0.0031$, Cohen d = 3.77 at 16.0 kHz, $p = 0.0034$, Cohen d = 4.72 at 22.6 kHz, and $p = 0.0005$, Cohen d = 4.37 at 32.0 kHz; Fig 4B). The proportion of synaptic loss was greatest at 5.6 kHz. Although we also counted orphan ribbons, which indicated ANF degeneration without the loss of presynaptic ribbons in the IHCs, no difference in the number of orphan ribbons was detected (Fig 4C). Analysis of auditory metrics and cochlear histology showed that the ABR P1 amplitude was statistically significantly correlated with the number of IHCs, OHCs, and synapses at several frequencies (IHC: 16.0 kHz: $p = 0.031$, r = 0.85, 22.6 kHz: $p = 0.0018$, r = 0.97, 32.0 kHz: $p = 0.0058$, r = 0.94 (Fig 3D); OHC: 16.0 kHz: $p = 0.033$, r = 0.85, 22.6 kHz: $p = 0.043$, r = 0.83, 32.0 kHz: $p = 0.012$, r = 0.91 (Fig 3F); synapse: 16 kHz: $p = 0.033$, r = 0.85, 22.6 kHz: $p = 0.039$, r = 0.83 (Fig 4D), simple linear regression test). The correlation analysis between synapse counts and ABR P1 amplitude at other frequencies demonstrated a trend toward proportional relationships (5.6 kHz: $p = 0.18$, 8 kHz: $p = 0.097$, 11.3 kHz: $p = 0.12$, 32 kHz: $p = 0.078$). Although we fitted a multiple linear regression model with ABR P1 amplitude as the dependent variable and IHC, OHC, and synapse counts as predictors, none of these predictors reached statistical significance at any frequency. Likewise, orphan ribbon count showed no significant correlation with ABR P1 amplitude at any frequency. Furthermore, there was no statistically significant correlation between the P1 latency and the number of IHCs, OHCs, synapses, or orphan ribbons (Figs 3E, 3G, and 4E).

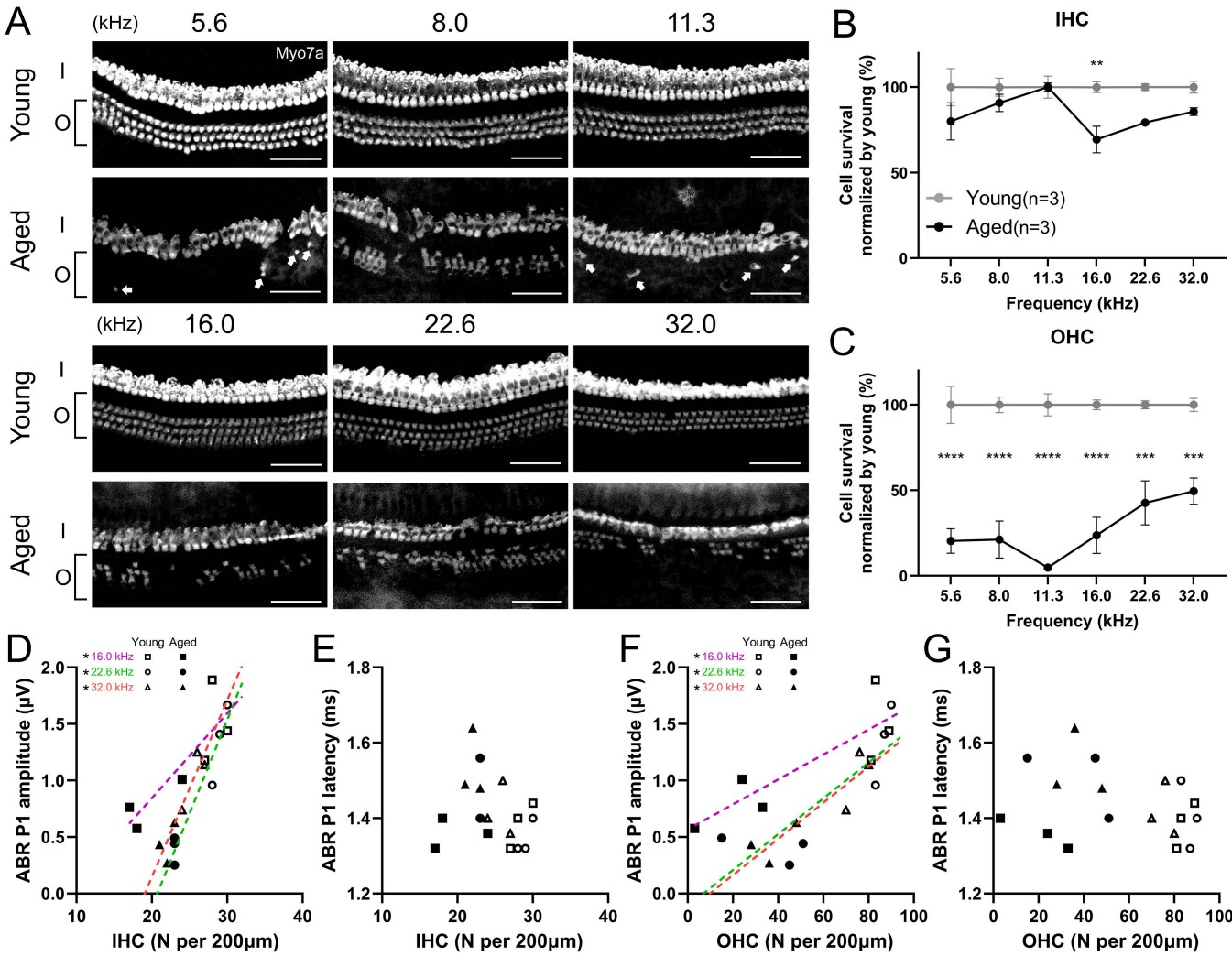

**Fig 3. Cochlear hair cell survival in aged and young mice.** (A) Representative immunofluorescence images of the HCs stained with Myo7a. Although IHCs are relatively preserved, OHC loss is severe, especially at 5.6 and 11.3 kHz, where white arrows indicate residual OHCs. Scale bars represent 50 μm. (B, C) The average HC survival rate is shown (n = 3 ears per group). Data is normalized by young data. IHC loss was significant at 16.0 kHz in the aged group. In contrast, OHC loss is significant at all frequencies in the aged group, with the peak at 11.3 kHz, and tends to be lower at 5.6 and 8.0 kHz. (D-G) The scatter plots with simple linear regression lines showing the relationship between the number of IHCs/OHCs and the ABR P1 amplitude/latency. IHC and OHC counts exhibit significant correlation to ABR P1 amplitude at 16.0, 22.6, and 32.0 kHz. No significant correlation is detected in P1 latency. Error bars represent standard error of mean. Asterisks indicate significant differences. $*p < 0.05$, $**p < 0.01$, $***p < 0.001$, $****p < 0.0001$. HC, hair cell; IHC/I, inner hair cell; OHC/O, outer hair cell; ABR, auditory brainstem response; ABR P1, ABR wave peak 1.

## Morphological changes in SM and the SV

Cross-sectional images were used to assess the morphological changes in the SV and SM, which were segmented into apical, middle, and basal turns (n = 5 ears in both groups). TEM images were used to observe detailed cellular morphological changes in SV.

 Cross-sectional images, including SGNs in the Rosenthal's canals, organs of Corti, SV, and spiral ligament, were stained with toluidine blue (Fig 5A). The SV displayed significant atrophy and thinning at all turns (Fig 5B). The average SV area of the aged group normalized by the young group was as follows: 0.57 at apical turn (6–8 kHz) (p = 0.0014, Cohen

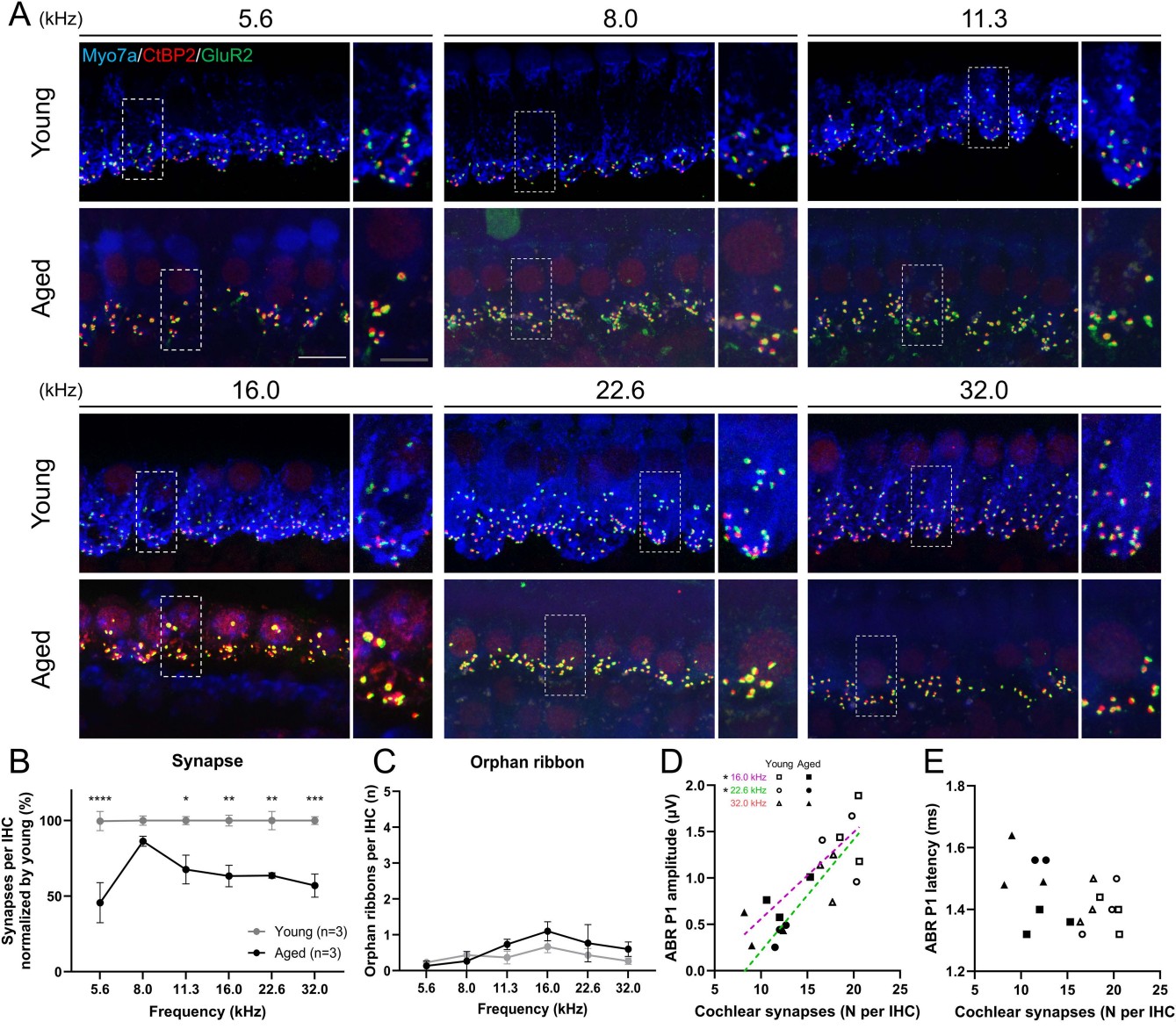

**Fig 4. Cochlear synapses in aged and young mice.** (A) Immunofluorescence images of synapses between IHCs and ANFs stained with CtBP2 (red) and GluR2 (green). The scale bars represent 10 µm (white) and 5 µm (gray), respectively. (B) The average count of synapses per IHC is shown (n = 3 ears per group). Data is normalized by young data. Synapses are significantly reduced, predominantly at 5.6 kHz in the aged group. (C) The average number of orphan ribbons per IHC is shown (n = 3 ears per group). No significant difference is observed in the number of orphan ribbons. (D, E) The scatter plots with simple linear regression lines showing the relationship between the number of synapses and ABR P1 amplitude and latency. A significant correlation is detected in P1 amplitude at 16.0 and 22.6 kHz. No significant correlation is found in P1 latency. Error bars represent standard error of mean. Asterisks indicate significant differences. *$p < 0.05$, **$p < 0.01$, ***$p < 0.001$, ****$p < 0.0001$. IHC, inner hair cell; ANF, auditory nerve fiber; ABR P1, auditory brainstem response wave peak 1.

d = 1.90), 0.62 at middle turn (12–16 kHz) ($p = 0.0049$, Cohen d = 2.47), and 0.63 at basal turn (24–32 kHz) ($p = 0.0061$, Cohen d = 3.56) (two-way ANOVA, interaction: F (2, 24) = 0.098, $p = 0.91$; Šidák post hoc test, respectively). Consistently, per-mouse SV area analysis showed statistically significant differences (S3 Fig and S4 Table). Although none of the turns showed a statistically significant difference in the SM area (Fig 5C), we observed a trend toward significance in the apical

turn with a large effect size (two-way ANOVA, interaction: F (2, 24) = 0.72, $p = 0.50$; Šidák post hoc test: $p = 0.052$, Cohen d = 2.05). Edematous changes were observed in the aged group, especially in the apical turn, which showed an extended Reissner's membrane.

In addition to the apparent atrophy and thinness of the SV, the aged group exhibited obstruction and narrowing of the capillary vessels accompanied by the accumulation of dark-stained masses in magnified images at all turns (Fig 6A). TEM images displayed the detailed morphology of the SV (Fig 6B and 6C), indicating the degeneration of intermediate cells with shrinkage of their process of adherence to marginal cells. In the younger group, the vessels were round. In contrast, the aged group exhibited distorted vessels, endothelial and basement membrane hypertrophy of the capillaries, and erythrocytes trapped in narrow intracapillary spaces. The dark-stained masses were hyperpigmented within the perivascular macrophages (PVM), indicating malfunction and the consequence of the tissue protective reaction against oxidative stress [22].

## Survival of SGNs and ANFs

To explore the neural pathology in the cochlea, we assessed SGNs and ANFs. Magnified cross-sectional TEM images were used for the SGNs. The staining intensity of the SGN cytoplasm was reduced, and the clustering of the cell bodies and axons was sparse in the aged group (Fig 7A). In the TEM images, SGN cell bodies exhibited several morphological changes. Vacuole formation, lipofuscin granule deposition, nucleoli separation, and mitochondrial malformation and depletion, including ballooning, were observed in the aged group (Fig 7B, C). Additionally, we performed quantitative

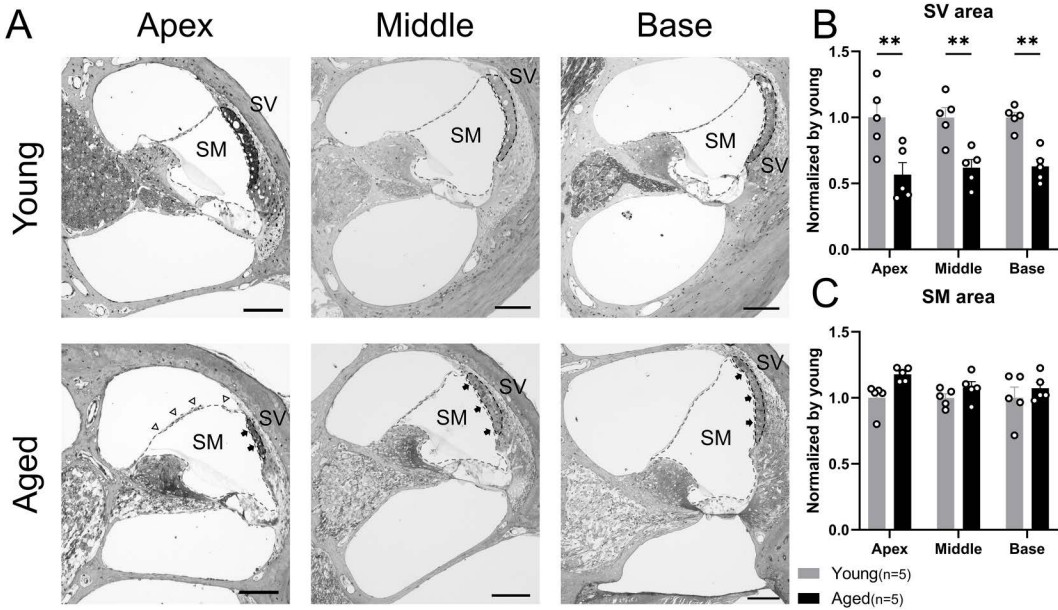

**Fig 5. Cross-sectional assessment of cochlear turn in aged and young mice.** (A) Cross-sectional images of apical, middle, and basal turns, including SGNs at Rosenthal's canals, the Organ of Corti, the SV (the area circled by black dashed line), and the spiral ligament are stained with toluidine blue. Black arrows represent the thinned SV compared with the young group. White arrowheads represent the extended Reissner's membrane due to the edematous change of the SM (the area circled by the gray dashed line), which is intense at the apical turn. Scale bars represent 100 μm. (B) The average SVA at each level of cochlear turns normalized to that in the young group (n = 5 ears per group). The SVA of all turns is significantly reduced, indicating significant atrophy and thinning of SV in the aged group. (C) The average SM area at each level of cochlear turn normalized to that of the young group is shown (n = 5 ears per group). The tendency for edematous change is observed with large effect sizes in the aged group, especially at the apical turn. Error bars represent standard error of mean. Asterisks indicate significant differences. **$p < 0.01$. SGN, spiral ganglion neuron; SV, stria vascularis; SM, scala media.

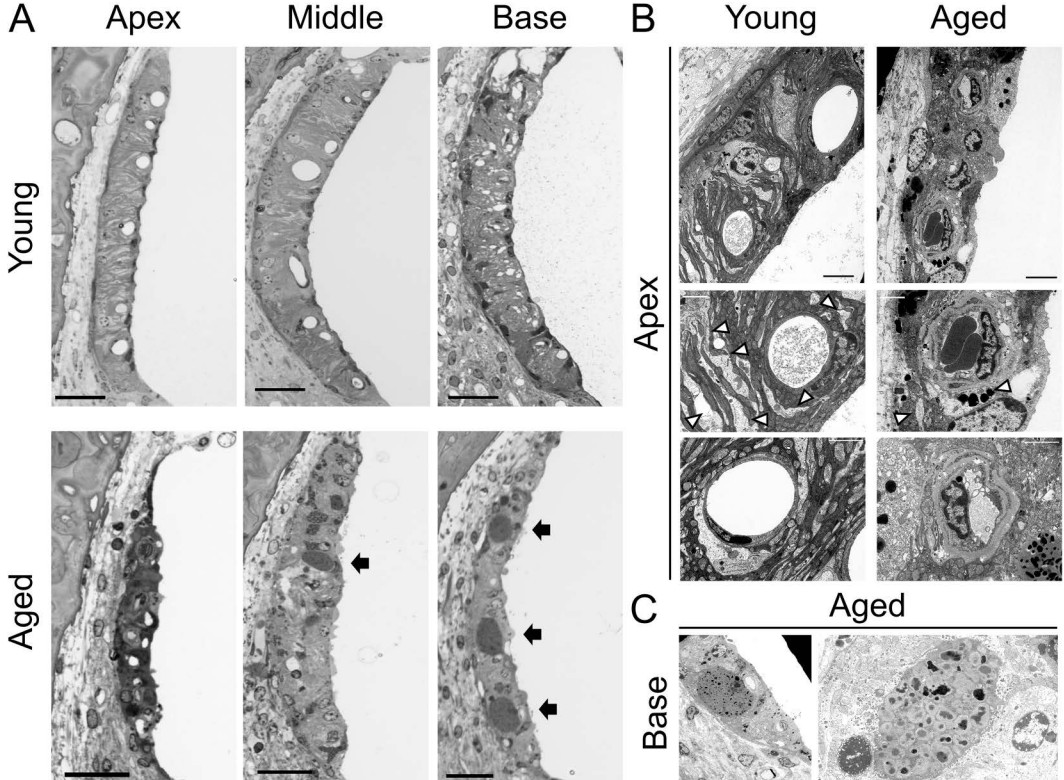

**Fig 6. Morphological change in stria vascularis of aged mice.** (A) Magnified SV images stained with toluidine blue from semithin sections plastic-embedded. Narrowing of capillary vessels is observed in the aged mice compared with the clearly round shapes in the young mice. Black arrows represent the accumulation of dark-stained masses in the aged SV. Scale bars represent 25 μm. (B) Representative TEM images of SV. The aged group exhibits degeneration of intermediate cells with shrinkage of their process adhering to marginal cells, distorted shape of capillaries with endothelial hypertrophy. White arrows indicate the representative processes of intermediate cells of each group. Black and white scale bars represent 5 μm and 2 μm, respectively. (C) Magnified TEM images of the dark-stained masses in SV, which indicate hyperpigmentation on macrophages. Black scale bars represent 2 μm. SV, stria vascularis; TEM, transmission electron microscope.

analysis of the average SGN density, cell size, and mitochondrial number (Fig 7D). The aged group showed a significant decrease in SGN density when compared to the young group (two-way ANOVA, interaction: $F_{(2, 24)} = 0.0048$, $p = 0.995$; Šidák post hoc test: $p < 0.0001$ at apical, middle, and basal turns, Cohen $d = 3.27$, 4.25, and 2.81, respectively, n = 5 ears in both groups). The percentages of survival SGN in the aged group compared to the young group were approximately 60% ($57 \pm 6.1$, $58 \pm 2.9$, $57 \pm 6.1$ at the apical, middle, and basal turn, respectively). The number of SGN mitochondria, as determined by TEM, was significantly reduced in all cochlear turns (two-way ANOVA, interaction: $F_{(2, 12)} = 0.077$, $p = 0.93$; Šidák post hoc test: $p = 0.0022$, Cohen $d = 5.44$ at the apical turn, $p = 0.0023$, Cohen $d = 3.05$ at the middle turn, and $p = 0.0054$, Cohen $d = 3.48$ at the basal turn, n = 3 ears in both groups). However, there were no significant changes in SGN cell size.

For ANFs, we assessed the peripheral axons in OSL (n = 3 ears in both groups). The density of axons in the aged mice was sparse (Fig 8A), with a significant decrease in the apical turn (two-way ANOVA, interaction: $F_{(1, 8)} = 1.27$, $p = 0.29$; Šidák post hoc test: $p = 0.013$, Cohen $d = 3.43$; Fig 8C). Considering the morphological changes observed in the TEM images, the aged mice had a reduced number of intra-axonal fiber structures, such as neurofilaments and microtubules (Fig 8B). Quantitative analysis using TEM images revealed a significant loss of mitochondria in the axons (two-way ANOVA, interaction: $F_{(1, 8)} = 0.062$, $p = 0.81$; Šidák post hoc test: $p = 0.018$, Cohen $d = 2.26$ at the apical turn (4–12 kHz),

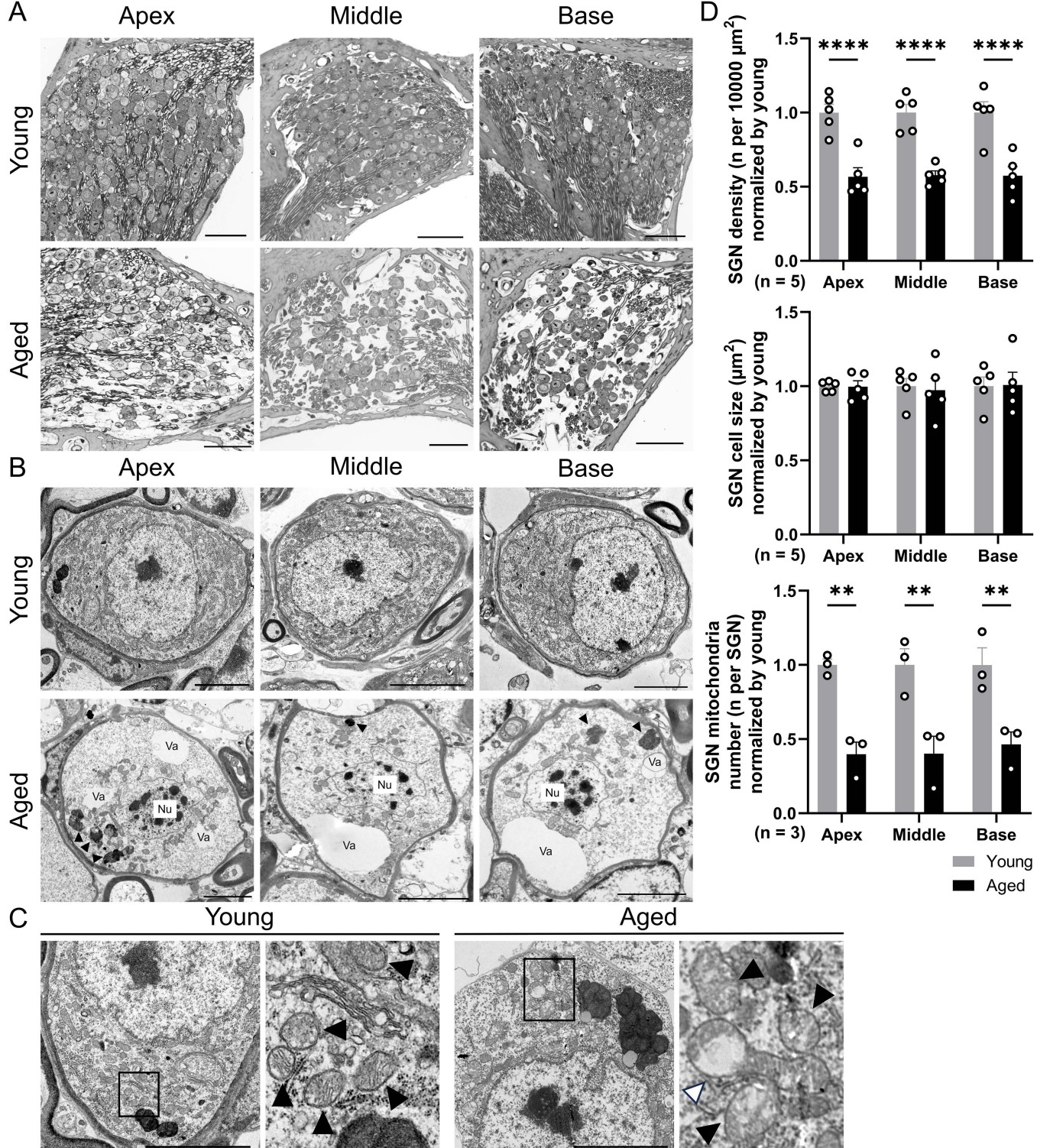

**Fig 7. Histopathology of spiral ganglion neurons in aged mice.** (A) Magnified SGN images stained with toluidine blue. The staining of the cell body is thin, and the density of the cell body and axon appears sparse. Scale bars represent 50 μm. (B) Representative TEM images of SGNs. The aged group displays vacuole formation (Va), lipofuscin granule deposition (black arrowheads), separated nucleoli (Nu), and mitochondrial malformations and depletion. Scale bars represent 5 μm. (C) Magnified TEM images of the SGN cytoplasm. Mitochondria are indicated by black arrowheads, and

"ballooning" of the outer mitochondrial membrane in the aged mice is indicated by a white arrowhead. Scale bars represent 5 μm. (D) The quantitative analysis of the average SGN density (n = 5 ears per group), cell size (n = 5 ears per group), and mitochondria number (n = 3 ears per group) is normalized to the young group. The aged group shows a significant decrease in SGN density and SGN mitochondria number compared with the young group. Error bars represent standard error of mean. Asterisks indicate significant differences. **$p < 0.01$, ****$p < 0.0001$. SGN, spiral ganglion neuron; TEM, transmission electron microscopy; Va, vacuole; Nu, nucleolus.

and $p = 0.011$, Cohen d = 4.43 at the basal turn (20–64 kHz); Fig 8C). Interestingly, none of the myelination indicators showed statistically significant differences (Fig 8D).

Regarding per-mouse analysis, we detected statistically significant differences for SGN density, SGN mitochondria number, axon density, and axon mitochondria number, which were the same as per-ear analysis (S3 Fig and S4 Table).

## Plasma d-ROM assessment

Given the numerous histological indicators of oxidative stress in aged mice, we assessed the serum level of d-ROMs, which reflects hydroperoxide (ROOH), a metabolite produced by systemic oxidative stress [23]. The average value of d-ROM was increased in the aged group when compared with that in the young group (204.2 ± 30.6 vs. 114.8 ± 7.9

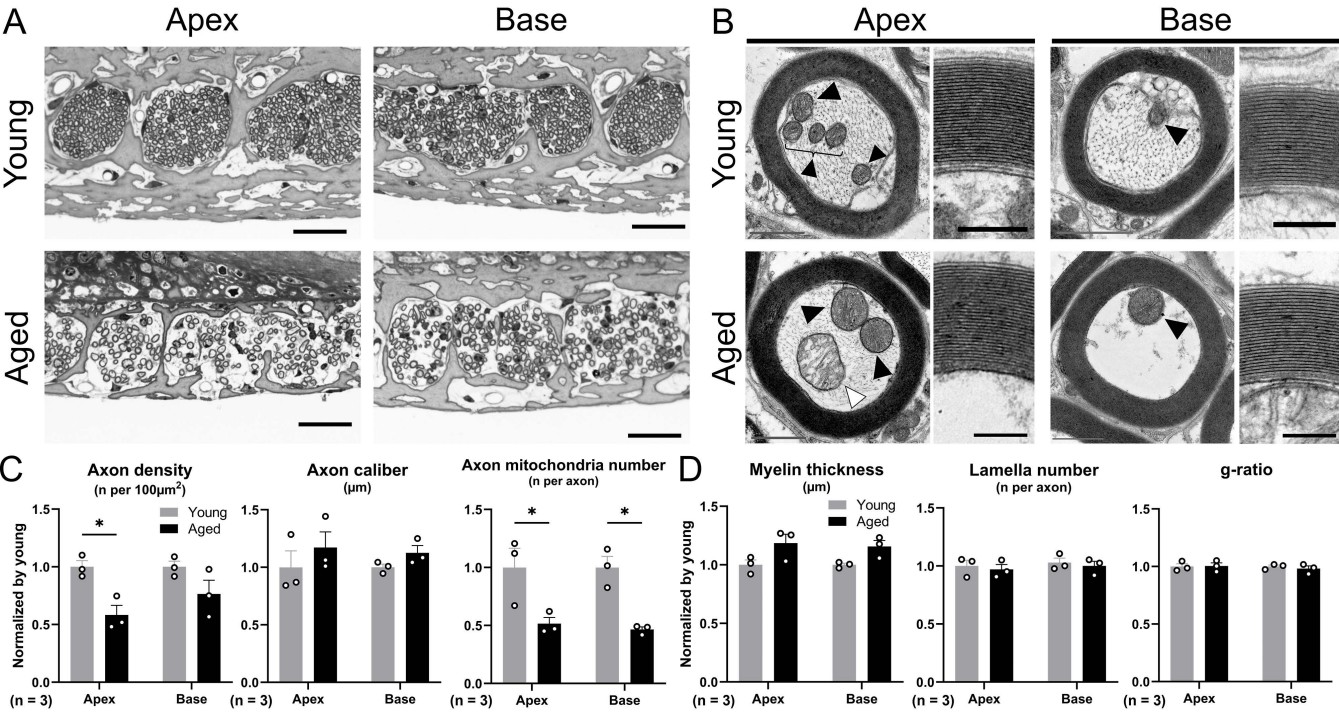

**Fig 8. Histopathology of peripheral axon and myelin of cochlear nerve in aged mice.** (A) Magnified images of peripheral axons of ANFs in OSL stained with toluidine blue. The density of axons in aged mice is sparse. Scale bars represent 25 μm. (B) Representative TEM images of axons and myelin. The rectangular images show the lamella of myelin. Neurofilaments and microtubules, which are intraaxonal fiber structures, are reduced in aged mice. Black arrowheads indicate the mitochondria in axons, and a white arrowhead in the aged group indicates a mitochondrion with disorganized and enlarged cristae. Gray and black scale bars represent 1 μm and 250 nm, respectively. (C) The quantitative analysis of the average of axon density, axon caliber, and axon mitochondria number, with values normalized to those of the young group is shown (n = 3 ears per group). The aged group shows a significant decrease in axon density and SGN mitochondria number when compared with the young group. (D) The quantitative analysis of the average myelin thickness, myelin lamella number, and g-ratio, with values normalized to those of the young group are shown (n = 3 ears per group). No indicators of myelination show statistical difference. Error bars represent standard error of mean. Asterisks indicate significant differences. *$p < 0.05$. ANF, auditory nerve fiber; OSL, osseous spiral lamina; TEM, transmission electron microscopy; SGN, spiral ganglion neurons.

(mean ± SEM), $p = 0.0079$, Cohen d = 1.79; Mann-Whitney $U$ test; Fig 9). Increased levels of d-ROMs indicate a disrupted balance between oxidative stress and the antioxidants in aged mice.

## Discussion

### Advance-aged auditory electrophysiological and cochlear histological changes

In the current study, we have elucidated the detailed pathophysiological changes in the cochlea of advanced-aged mice that survived under normal rearing conditions for more than 2.5 years. Various studies on age-related changes in the cochlea have been conducted using the human temporal bone and animal models. The three main pathological classifications by Schuknecht et al. have long been referred to as sensory presbycusis (loss of HCs), neural presbycusis (loss of cochlear neurons), and strial presbycusis (atrophy of SV) [7,8]. Notably, the advanced-age cochleae evaluated in this study showed a combined histology of all age-related changes. Therefore, the intense ABR threshold shift can be attributed to the loss of nonlinear vibration amplification mechanisms due to the loss of OHCs [24,25], EP reduction due to SV atrophy and subsequent impairment of HC function [9,10], and the loss of SGNs. This multifocal cochlear pathology is distributed throughout the cochlea and is associated with a significant threshold shift across a broad frequency range. However, several studies have reported that severe SGN loss is associated with a hearing threshold shift. Schuknecht et al. demonstrated that hearing threshold elevation was observed only when the SGN loss exceeded 75% [26]. Frisna et al. detected a moderate correlation between SGN loss and the hearing threshold in an aged mouse model, which required approximately 70% loss of SGNs [27]. These studies indicate that the 60% loss of SGNs in the current study would not contribute to the ABR threshold shift. Therefore, an elevated ABR threshold may primarily reflect HC loss and SV atrophy, as these cochlear pathologies were reported to be related to threshold shifts [28,29]. Moreover, given the absence of any apparent structural changes contributing to the sound transfer function in the tympanic membrane and ossicular chain during the histological evaluation, the elevated DPOAE threshold could be caused by the loss of OHCs, as implied in a previous animal study [30].

ABR P1 reflects the sum of cochlear neuron activity, whose amplitude depends on the number of activated neurons and synchrony of firing with intact HCs [31–33], and whose latency depends on the timing of synaptic transmission and nerve conduction related to myelination [33,34]. The statistically significant correlation between the number of synapses

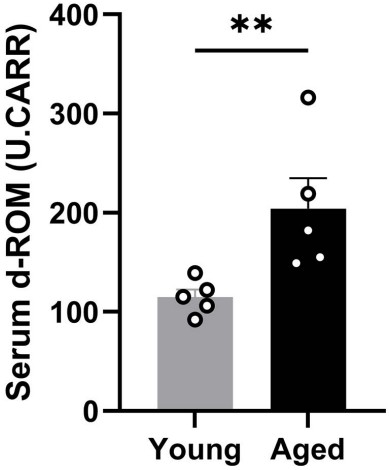

**Fig 9. Plasma d-ROM assessment.** The histogram shows the average d-ROM value in the plasma, representing systemic oxidative stress. The aged group has a significantly higher score than the young group (n = 5). Asterisks indicate significant differences. **$p < 0.01$. d-ROM, derivatives-reactive oxygen metabolite; U.CARR, Carratelli unit.

and the ABR P1 amplitude was detected at 16.0 and 22.6 kHz in the current study (Fig 4D), which was inconsistent with a previous report demonstrating significant correlation at all measured frequencies [11]. However, the scatter plots at other frequencies suggested a trend toward proportional relationships. We therefore suspect that the absence of intermediate-aged cohorts and the limited sample size may have reduced the statistical power to detect significant correlations. Similar to the ABR results in aged human beings [35], the aged mice in the current study exhibited significantly lower ABR P1 amplitude and latency prolongation than young mice (Fig 2C, D). A significant decrease in the number of SGNs was observed in the aged group, which may have resulted in the lower P1 amplitude. Zhang et al. investigated the link between SGN quantity and ABR P1 amplitude in mice and revealed a significant correlation that supports this hypothesis [31]. The amplitude of ABR P1 is also attributed to the loss of synapses between IHCs and peripheral ANFs of bipolar SGNs [36], with this reduction combined with the ABR P1 amplitude reduction in advanced-aged mice. Furthermore, the loss of HCs also affects ABR P1 amplitude. In a chinchilla model, carboplatin-induced IHC loss, accompanied by relatively minor OHC loss, resulted in a reduction of the compound action potential amplitude [37]. OHC loss or functional deficit also independently showed a lower ABR P1 amplitude in mouse models [38,39]. We also demonstrated the significant correlation between IHC/OHC counts and P1 amplitude. Conclusively, combined histological damage in the cochlear components is likely to affect low ABR P1 amplitude in advanced-aged mice. Regarding the ABR P1 latency, although nerve conductive velocity was generally correlated with axonal myelination, no statistically significant difference was observed in any myelination indicator between the aged and younger groups (Fig 8D). Furthermore, no significant correlation was observed between the number of synapses and the P1 latency. Therefore, the factor affecting latency prolongation in advanced-aged mice could be the dispersed timing of synaptic transmission related to the reduced number of synapses and ANFs, rather than axonal demyelination.

Orphan ribbons are synaptic ribbons that persist after ANF terminations have retracted, and their numbers are substantially increased by acute glutamate excitotoxicity due to noise exposure or blast injuries [15,40]. A previous study demonstrated that the number of orphan ribbons was elevated in the basal turns of aged CBA/CaJ mice [41]. By contrast, we found no difference in the number of orphan ribbons per IHC between the aged and young groups. We assumed that older age could be one of the factors causing this inconsistency, as another study reported that presynaptic ribbons progressively disappeared after postsynaptic density degeneration [42].

Cross-sectional histological assessments of the cochlea revealed an increased SM area and expansion of the Reisner's membrane, indicative of endolymphatic hydrops in the aged group, especially at the apical turn, with a large effect size (Fig 5). Various ion channels and transporters participate in the maintenance of endolymph ion composition [43], and aquaporin channels have been speculated to play an essential role in the etiology of hydrops [44,45]. However, the mechanisms underlying the development of endolymphatic hydrops remain unclear. Although age-related changes are generally considered to progress from the basal turn, a recent detailed study reported that in humans and mice, OHCs begin to fall away from both basal and apical turns with age, subsequently progressing to the entire cochlea, including the middle turn [11,46,47]. Therefore, the one of unknown etiology related to loss of HCs in the apical turn of aged cochleae may involve endolymphatic hydrops we observed as an age-related change.

### The impact of environmental sound on cochlear aging

As mentioned above, one of the histologic manifestations of age-related hearing loss is HC loss, particularly the loss of OHCs that precede IHCs [48], occurring in both the basal and apical turns and extending across the cochlea [11,46,47]. The most significant OHC degeneration observed in the current study was in the mid-frequency cochlear region. Furthermore, the intensity of the OHC loss was found to be greater at lower frequencies than at higher frequencies. We hypothesized that the accumulation of acoustic stimuli from steady environmental sounds would affect the outcome of OHC loss. Previous studies have suggested that steady-state sub-traumatic sound exposure may affect age-related hearing loss, given that reduced ambient sound can mitigate hearing threshold shifts or HC loss. Rosen et al. reported a slight increase

in pure-tone hearing thresholds among older individuals in a Sudanese tribe, which was located in an isolated primitive area with ambient sound levels below 40 dB [49]. Tarnowski et al. have reported that gerbils aged in a quiet vivarium (<41 dB A) retained IHCs and OHCs comparable to young gerbils [50]. Considering that our animal care facility had a relatively high background sound level (LZeq > 70 dB SPL) and that previous reports have shown suppressed OHC loss in a quiet environment, we speculated that the age-related OHC loss in the current study could be attributed to the exposure of mice to steady ambient sound for more than 2.5 years. These findings suggest that chronic exposure above ~40 dB SPL may contribute to age-related hearing loss, whereas lower levels may help preserve auditory function. Moreover, the frequency characteristics of steady environmental sounds could be critical in relation to the background sounds and OHC loss in this study. Animal models exposed to octave or narrowband noise exhibit OHC loss corresponding to the exposed noise [51,52]. Therefore, it is likely that the HC damage caused by noise exposure correlates with the frequency distribution. The high proportion of the ambient sound pressure in our vivarium (Fig 1) was dominated by lower frequencies (5.6, 8.0, and 11.3 kHz) within the audible range of mice (1–100 kHz) [53]. Consequently, the accumulation of acoustic stimulation induced by background sound may constitute a potential factor contributing to cochlear damage, which can result in age-related hearing loss. However, it is essential to acknowledge that the current study is observational and lacks an interventional component, which limits its ability to demonstrate a definitive and unequivocal causal relationship.

Human temporal bone pathology and mouse experiments have revealed that synaptic connections between IHC and ANF are the most vulnerable to acoustic stimuli [36], and age-related loss of synapses precedes the loss of HCs or SGNs [11,12]. We observed a dominant loss of synapses and peripheral lateral axons in the cochlear nerve at low frequencies (Figs 4 and 8). As mentioned previously, the ambient sound of the vivarium in this study had a solid SPL corresponding to the low-frequency range of audible frequencies in mice. Feng et al. demonstrated more severe synaptic loss in the region of 30–70% of the cochlear turn in mice exposed to low-intensity white noise for a long duration [54], suggesting that this audible spectrum of mice, corresponding to 12–32 kHz [20], is vulnerable to prolonged acoustic exposure related to synaptic degeneration. In this regard, the ambient sound measured in the current study exhibited a frequency profile with a peak shifted toward lower frequencies; thus, the synaptic loss may have been more pronounced at 5.6 kHz. Considering our results together with previous studies, the accumulation of acoustic stimuli in combination with aging may contribute to the reduction of synapses and auditory nerve fibers (ANFs). However, the rationale for the preservation of synapses specifically in the 8.0 kHz region remains unclear and may involve unknown pathological mechanisms or combined factors such as individual variability, strain, sex, or rearing conditions, aside from ambient sound. This uncertainty should be acknowledged as a limitation of the present study and warrants further investigation.

### The influence of oxidative stress on aging SGNs

The loss of SGNs occurs via two mechanisms: primary neural degeneration, in which synapses, axons, and neural bodies degenerate first, and secondary degeneration, in which neural loss is preceded by the loss of IHCs and supporting cells. Primary neural degeneration is considered an important mechanism in age-related changes, as both human temporal bone pathology and animal studies have revealed that age-related SGN loss is more severe than IHC loss, following synapse and ANF loss [11,12,55]. A study assessing age-related cochlear changes in mice and superoxide dismutase (SOD), a protective enzyme against reactive oxygen species (ROS)-induced oxidative stress, suggested a link between oxidative stress and the loss of SGNs [56]. In that study, SOD1 knockouts showed an early ABR threshold shift and a decrease in the number of SGNs. Furthermore, the accumulation of ROS-induced mitochondrial DNA damage and the resulting functional decline in mitochondria are important factors in age-related SGN loss, particularly in primary neural degeneration [57]. In the present study, an even level of SGN loss was observed from the basal to the apical turn. The percentage of SGN survival was approximately 60%, which was lower than that previously reported for the same mouse strain and age (approximately 130-week-old) [11], although the percentage of IHC loss was similar. TEM histology revealed findings suggestive of cellular senescence, such as lipofuscin granule deposition, intracellular vacuolation, and segmented nucleoli in

the SGN soma (Figs 7 and 8). In particular, lipofuscin reportedly functions as an indicator of intracellular oxidative stress, given that its accumulation is increased by oxidative stress [58], and its binding to iron ions enhances ROS production and further aggravates oxidative stress [59]. Lipofuscin granules accumulate in the cell bodies of various cells in the inner ear with age [60], and a study using senescence-accelerated prone mice reported substantial accumulation of lipofuscin granules in SGNs [61]. Lipofuscin granule accumulation was also observed in the SGNs of naturally advanced aged mice, indicating that SGNs were under severe oxidative stress.

Furthermore, age-related deterioration of mitochondrial function and the accumulation of mitochondrial DNA mutations may enhance oxidative stress, resulting in a vicious cycle that induces apoptosis and necrosis, ultimately leading to cell loss in the cochlea and age-related hearing loss [62]. We also found a decrease in the number of mitochondria and morphological changes in SGNs and ANFs, such as mitochondrial ballooning and damaged cristae, indicating severe mitochondrial dysfunction and oxidative stress [63,64]. Accordingly, primary degeneration in the cochlear neurons of advanced-aged mice is potentially enhanced by metabolic factors, resulting in a high degree of cell loss.

Similarly, histological findings suggest that oxidative stress may influence age-related changes in the SV. In the current study, severe SV atrophy was observed, and morphological findings such as lumen deformation and narrowing, thickening of the basement membrane in capillary vessels, and melanin pigmentation in the PVM were observed in the microstructure. These findings are similar to those reported previously, although the animal species and strain differed [65,66], indicating that these findings are common to age-related SV changes. Recent studies have indicated that oxidative stress may increase PVM activation and the associated pigmentation [67,68]. Therefore, it is likely that increased oxidative stress affects age-related tissue changes in the SV.

We also measured d-ROM levels as a marker for assessing oxidative stress in advanced-aged mice. d-ROMs reflect the hydroperoxide (ROOH) produced by oxidative stress and can be used to assess the systemic oxidative stress status [23]. Furthermore, histological deformations were observed in the current study, including a reduction in the mitochondrial number in the SGN soma and axon, as well as mitochondrial malformations such as ballooning and damaged cristae. These findings are related to the high level of oxidative stress in the cochlea, which has been reported in an enzyme-deficient mouse model, inducing mitochondrial tRNA dysfunction and activating the apoptotic pathway [69,70]. Serum d-ROM levels were found to be significantly elevated in the aged group, suggesting the presence of excess oxidative stress, which is potentially associated with the histological findings reflecting oxidative stress and mitochondrial damage.

## Limitations

Although we performed a comprehensive electrophysiological and histomorphological assessment in naturally senescent advanced-aged mice, a complete evaluation could not be performed. A limitation of this study is that the lowest ABR threshold was set at 20 dB SPL, which may have restricted the evaluation of potential lower thresholds, particularly in the younger group. Regarding electrophysiological aspects, we could not measure or evaluate EP, which could lead to low-frequency hearing loss. However, several studies have reported SV atrophy, vascular degeneration, and reduced ion channel expression [9,71,72], which can affect the EP reduction [73]. Therefore, our findings of SV atrophy, as observed by light microscopy and TEM, suggest a potential decrease in EP.

Another limitation is the lack of a middle-aged mouse group. Although our primary objective was to characterize the cochlear degeneration in advanced-aged CBA/CaJ mice, the inclusion of a middle-aged cohort would have allowed us to accurately determine the onset and progression of age-related changes in cochlear histopathology. This limitation prevents the establishment of a critical window during which preventive or therapeutic interventions may be most effective.

Morphological evaluation of SGNs suggested that oxidative stress and mitochondrial damage may have induced SGN loss; however, this study did not perform a functional assessment. Furthermore, we were unable to perform correlation analyses for this dataset because the d-ROM data and results of auditory function/cochlear histology were obtained from different individuals. Oxidative stress and mitochondrial dysfunction can be directly assessed

by evaluating the activities of enzymes such as SOD and glutathione peroxidase, well-known antioxidant enzymes, and cytochrome oxidase, which is necessary for mitochondrial cellular respiration, and have been explored previously, including in basic research [62,74,75]. In this study, we examined age-related morphological, cellular, and tissue changes in the cochlea. Therefore, we assumed that the antioxidant capacity and mitochondrial function were also reduced.

Our small sample size and incomplete independence within the group in the cochlear histological analysis are other limitations. Restrictions on the domestic distribution of the same strain in our country, combined with the difficulty in maintaining mice near the end of their lifespan, necessitated the use of this small sample size. Additionally, to adhere strictly to ethical principles and minimize the number of animals sacrificed, we utilized both ears from a single animal within the same evaluation method and group rather than doubling the number of animals sacrificed. However, the calculated sample size to yield a power of > 0.8 to detect significant differences with an alpha of 0.05 showed that the number of specimens we adopted was acceptable in terms of the survival percentage of IHC, OHC, synapse, and SGN in the advanced-aged group, as cited from a previous study using the same strain [11]. Thus, we consider our histological results worthy of discussion regarding advanced-aged cochlear histomorphological deterioration.

## Conclusions

In this study, we performed a comprehensive electrophysiological and histomorphological evaluation of the auditory function and cochlear tissues in advanced-aged CBA/CaJ mice in a normal feeding environment under steady ambient sound. We found that the loss of OHCs and synapses, which are considered vulnerable to acoustic stimuli, predominantly occurred in the responsible frequency region, which corresponds to the frequency characteristics of environmental sounds. This suggests that the long-term accumulation of acoustic stimuli may contribute to the pathogenesis of age-related hearing loss, even in the absence of intense environmental sounds. Severe loss of SGNs in advanced-aged mice may occur due to primary neural degeneration, and the coexistence of mitochondrial damage and systemic oxidative stress indicates metabolic dysfunction in SGNs. The accumulation of acoustic and metabolic damage could be a potential factor in the occurrence and progression of age-related hearing loss. Therapeutic or preventive interventions for these factors may reduce the impact of age-related hearing loss in a rapidly aging society.

## Supporting information

**S1 Fig. Cochlear electrophysiological function results per mouse in the aged and young groups.** (A) DPOAE thresholds. (B) ABR thresholds. (C) ABR P1 amplitude. (D) ABR P1 latency. The number of subjects is 5 mice per group. The overall results are consistent with per-ear comparing analysis. Error bars represent standard error of mean. Asterisks indicate significant differences. $*p < 0.05$, $**p < 0.01$, $***p < 0.001$, $****p < 0.0001$.
(TIF)

**S2 Table. Two-way ANOVA and post hoc test results of per-mouse cochlear electrophysiological function test in aged and young mice.**
(DOCX)

**S3 Fig. Cochlear cross-section and TEM histological quantitatively analyzed results per mouse in the aged and young groups.** (A) SV area. (B) SM area. (C) SGN density. (D) SGN cell size. (E) SGN mitochondria number. (F) Axon caliber. (G) Axon density. (H) Axon mitochondria number. (I) Myelin thickness. (J) Lamella number. (K) g-ratio. The number of subjects is listed in S4 Table. The overall results are consistent with per-ear comparing analysis. Error bars represent standard error of mean. Asterisks indicate significant differences. $*p < 0.05$, $**p < 0.01$, $***p < 0.001$.
(TIF)

**S4 Table. Two-way ANOVA and post hoc test results of per-mouse cochlear cross-section and TEM analysis in aged and young mice.**
(DOCX)

**S1 File. Source data for Fig 1.**
(CSV)

**S2 File. Source data for Fig 2.**
(CSV)

**S3 File. Source data for Fig 3.**
(CSV)

**S4 File. Source data for Fig 4.**
(CSV)

**S5 File. Source data for Fig 5.**
(CSV)

**S6 File. Source data for Fig 7.**
(CSV)

**S7 File. Source data for Fig 8.**
(CSV)

**S8 File. Source data for Fig 9.**
(CSV)

## Acknowledgments

We would like to thank Prof. Ken Adachi (Department of Cardiology, National Defense Medical College) for measuring the d-ROMs and for the valuable discussions. We also wish to thank Yayoi Ichiki (a technical expert, Central Research Laboratory, National Defense Medical College) for providing research assistance.

## Author contributions

**Conceptualization:** Kunio Mizutari, Jun Suzuki.

**Data curation:** Yoshiaki Inuzuka, Takaomi Kurioka.

**Formal analysis:** Yoshiaki Inuzuka, Takaomi Kurioka.

**Funding acquisition:** Kunio Mizutari, Takaomi Kurioka, Akihiro Shiotani.

**Investigation:** Yoshiaki Inuzuka.

**Methodology:** Takaomi Kurioka, Jun Suzuki, Yutaka Koizumi.

**Project administration:** Kunio Mizutari.

**Resources:** Jun Suzuki, Yutaka Koizumi, Akihiro Shiotani.

**Supervision:** Kunio Mizutari, Koji Araki, Akihiro Shiotani.

**Validation:** Yoshiaki Inuzuka, Takaomi Kurioka.

**Visualization:** Yoshiaki Inuzuka, Takaomi Kurioka, Yutaka Koizumi.

**Writing – original draft:** Yoshiaki Inuzuka, Kunio Mizutari, Takaomi Kurioka.

**Writing – review & editing:** Kunio Mizutari.

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
