## [Editor Report · Decision Letter 0]

30 Apr 2025

PONE-D-25-21282Histopathological change of age-related hearing loss in advance-aged CBA/CaJ micePLOS ONE

Dear Dr. Mizutari,

Thank you for submitting your manuscript to PLOS ONE. After careful consideration, we feel that it has merit but does not fully meet PLOS ONE’s publication criteria as it currently stands. Therefore, we invite you to submit a revised version of the manuscript that addresses the points raised during the initial review by the Editor.

We look forward to receiving your revised manuscript.

Kind regards,

Miriam A Hickey, PhD

Academic Editor

PLOS ONE

2. To comply with PLOS ONE submissions requirements, in your Methods section, please provide additional information regarding the experiments involving animals and ensure you have included details on (1) methods of sacrifice, and (2) efforts to alleviate suffering.

“This work was supported by a grant for Advanced Defense Medical Research provided by the Japanese Ministry of Defense (Grant Number A-4, A.S.) and two grants from JSPS KAKENHI (Grant Numbers 21K09573, K.M. and 20K18263, T.K.).”

Editor Comments

The old group of mice appears to comprise nine female CBA/CaJ mice 104 aged between 129 and 138 weeks whereas the young group appears to comprise seven male CBA/CaJ mice aged 9-14 weeks.

Due to the established sex-based differences in hearing and hearing loss in males versus females (humans and mice), the authors are asked to provide a sex-matched young group, for comparisons.

Reviewers' comments:

No external Reviewer comments thus far. This is the initial review by the Editor.

---

## [Author Response · Author response to Decision Letter 1]

6 Jun 2025

Dear Dr. Hickey,

Thank you very much for your careful review and the helpful feedback provided during the initial check. We have carefully revised the manuscript in accordance with your suggestions and are submitting the updated version for your further consideration. Please find below our point-by-point responses to the issues raised.

Journal Requirements:

1. Formatting:

“Please ensure that your manuscript meets PLOS ONE's style requirements, including those for file naming.”

We apologize for not using the correct formatting in our original submission. The manuscript has been reformatted entirely to follow the PLOS ONE style requirements.

2. Animal Experimentation Details:

“To comply with PLOS ONE submissions requirements, in your Methods section, please provide additional information regarding the experiments involving animals and ensure you have included details on (1) methods of sacrifice, and (2) efforts to alleviate suffering.”

In accordance with your guidance, we have revised the Materials and Methods section to include (1) the method of euthanasia and (2) efforts made to minimize animal suffering. Thank you for bringing this to our attention.

3. Funding Disclosure:

“Please state what role the funders took in the study. If the funders had no role, please state: "The funders had no role in study design, data collection and analysis, decision to publish, or preparation of the manuscript.”

As the funders had no role in the study design, data collection and analysis, decision to publish, or preparation of the manuscript, we have added the following statement to the cover letter:

Editor’s Comment on Sex Matching in Experimental Animals:

“The old group of mice appears to comprise nine female CBA/CaJ mice 104 aged between 129 and 138 weeks whereas the young group appears to comprise seven male CBA/CaJ mice aged 9-14 weeks. Due to the established sex-based differences in hearing and hearing loss in males versus females (humans and mice), the authors are asked to provide a sex-matched young group, for comparisons.”

We sincerely apologize for the incorrect description regarding the sex of the animals used in our study. The original manuscript incorrectly stated that the young group consisted of male mice, while the aged group consisted of females. In fact, both groups were composed of female CBA/CaJ mice. This inconsistency was a serious oversight on our part. We have corrected the manuscript accordingly. We greatly appreciate your careful review and the opportunity to rectify this error.

Please do not hesitate to contact us if any further clarification is needed. We thank you again for your attention and support of our work.

Sincerely,

Kunio Mizutari, M.D., Ph.D.

---

## [Decision Letter · Decision Letter 1]

6 Jul 2025

PONE-D-25-21282R1Histopathological change of age-related hearing loss in advance-aged CBA/CaJ micePLOS ONE

Dear Dr. Mizutari,

Thank you for submitting your manuscript to PLOS ONE. After careful consideration, we feel that it has merit but does not fully meet PLOS ONE’s publication criteria as it currently stands. Therefore, we invite you to submit a revised version of the manuscript that addresses the points raised during the review process.

Please address the comments of each Reviewer; the comments are available below.

We look forward to receiving your revised manuscript.

Kind regards,

Miriam A. Hickey, PhD

Academic Editor

PLOS ONE

Reviewers' comments:

Reviewer's Responses to Questions

**Comments to the Author**

1. If the authors have adequately addressed your comments raised in a previous round of review and you feel that this manuscript is now acceptable for publication, you may indicate that here to bypass the “Comments to the Author” section, enter your conflict of interest statement in the “Confidential to Editor” section, and submit your "Accept" recommendation.

Reviewer #1: All comments have been addressed

Reviewer #2: All comments have been addressed

Reviewer #3: (No Response)

Reviewer #4: (No Response)

Reviewer #5: (No Response)

2. Is the manuscript technically sound, and do the data support the conclusions?

Reviewer #1: Yes

Reviewer #2: Yes

Reviewer #3: Partly

Reviewer #4: No

Reviewer #5: Yes

3. Has the statistical analysis been performed appropriately and rigorously? 

Reviewer #1: Yes

Reviewer #2: Yes

Reviewer #3: Yes

Reviewer #4: No

Reviewer #5: Yes

4. Have the authors made all data underlying the findings in their manuscript fully available?

Reviewer #1: Yes

Reviewer #2: Yes

Reviewer #3: Yes

Reviewer #4: Yes

Reviewer #5: Yes

5. Is the manuscript presented in an intelligible fashion and written in standard English?

Reviewer #1: Yes

Reviewer #2: No

Reviewer #3: Yes

Reviewer #4: Yes

Reviewer #5: Yes

6. Review Comments to the Author

Reviewer #1: This study examined cochlear pathology and auditory function in aged CBA/CaJ mice, which maintain normal hearing when young. Compared to young mice, aged mice showed outer hair cell and synapse loss in frequency regions matching environmental noise exposure. Additional findings included stria vascularis atrophy, microthrombi, endolymphatic hydrops, and neural degeneration. Elevated oxidative stress markers were also observed. The results suggest ARHL arises from both chronic noise-induced damage and metabolic dysfunction due to mitochondrial decline and oxidative stress, highlighting the need to target both mechanisms for effective prevention and treatment. This study is interesting and valuable, however there are several concerns to publish in current version.

Major concerns

1. I believe this is a study comparing young and super-aged CBA/CaJ mice. However, if middle-aged mice were also included, it might have been possible to determine where age-related changes begin within the cochlea. Please add this point to the limitations section.

2. While it has been pointed out that mitochondrial function and metabolism within the cochlea are important, the discussion of systemic oxidative stress alone may weaken the argument.　There are several papers that have measured actual cochlear metabolism, so please cite them and add them to the discussion.

3. Authors concludes that long-term acoustic exposure is a cause of age-related hearing loss, but what level of acoustic exposure is considered acceptable? Would hearing be better preserved without acoustic exposure? Please discuss previous reports on long-term acoustic exposure.

Minor concerns

In Figures 7 and 8, the number of mitochondria is evaluated. How were the findings and evaluation of ballooning? Since it is unclear which structures are mitochondria, please mark them with arrows or other symbols.

Reviewer #2: Inuzuka and colleagues investigated hearing changes and cochlear morphological alterations in aged (129-138 weeks old) CBA/CaJ mice, providing laboratory data supporting the CBA/CaJ mouse as a reliable animal model for age-related hearing loss (ARHL). The experimental methods employed in this study are appropriate, and the results generally support the conclusions. However, the following issues need to be addressed prior to consideration for publication:

1. The sample size for cochlear histological analysis (n=3) is insufficient and must be increased to at least n>5 to ensure reliable and statistically robust data.

2. The observed significant atrophy of the stria vascularis in aged CBA/CaJ mice raises the question of whether this morphological change affects the endocochlear potential (EP). The authors should discuss this potential implication or provide relevant data if available.

3. The study measured reactive oxygen species (ROS) levels only in blood. Measuring ROS levels specifically within the cochlear tissue is crucial, particularly given that blood ROS levels may not accurately reflect ROS levels within the target organ/tissue. The authors should justify this omission or include cochlear ROS data.

4. The study exclusively used female mice. Consequently, the findings may not be representative of male animals. The title of the manuscript should explicitly state that the study was conducted on female mice (e.g., "...in Female CBA/CaJ Mice").

5. In all figures presenting comparative data (young vs. aged), the data for young animals should be placed before the data for aged animals. This arrangement provides a more logical progression for readers.

6. The manuscript requires thorough editing by a native English-speaking professional. Numerous grammatical and spelling errors are present. For example, on line 269, "10 ears of both groups…" should be written as "Ten ears…", consistent with standard English writing conventions for numbers at the beginning of a phrase. Careful proofreading is essential to improve clarity and professionalism.

Reviewer #3: Summary:

The manuscript “Histopathological change of age-related hearing loss in advance-aged CBA/CaJ mice” focuses on the important task of fully characterizing the pathology that may underly age-related hearing loss in an aging mouse model. The author’s use of advanced-age CBA/CaJ mice and multiple histopathological measurements in the same mice is a strength of this study. However, the authors failed to do any analysis associated the histopathological measurements with each other or directly with auditory functional measurements. This is a hugely missed opportunity, as without a direct association between audiological functioning and structure in the same animal, these results cannot draw any strong conclusions about the underlying drivers of age-related hearing loss. Furthermore, the authors make strongly worded conclusions about environmental noise and oxidative stress that are not fully supported by the data or appropriate literature review.

General:

1) One of the major conclusions by the authors is that environmental noise may be a driving factor in the underlying pathological changes in aged mice. I do not believe that this finding is fully supported by the data. Data from OHCs show that OHC loss was greatest at the low frequency regions of the cochlea, but also that there was severe OHC loss at the 16-32 kHz regions as well. Synapse loss was greatest at the 5.6 kHz region, but was fully recovered at 8 kHz with similar levels of synapse loss at the 11-32 kHz frequency range. Auditory thresholds were not the most affected at low frequencies and exhibited general hearing loss. For wave I analysis, 5.6 kHz exhibited no significant decrease in amplitude or latency in older mice. The effect of environmental noise is discussed only in the context of OHC, axon density, and synapse loss without discussion of the numerous other metrics analyzed in this study.

2) The authors missed the opportunity to compare across metrics in the same cohort of mice. For example, audiometric threshold is not compared with any structural measure in this study. It would be interesting to see how in the aged mice, number of synapses correlated with wave I amplitude or latency. Furthermore, why were ABRs only collected from 5 mice in each group? How many of the ears for histology had audiometric data associated with them? Without the study of associations between metrics, the conclusions of this study are not strong.

3) When reporting the statistics, please report the results of the 2-way ANOVA, not just the post-hoc Sidak test. Before reporting the Sidak test, please report the F and values for the interaction between predictor variables at minimum. Without this information, it is impossible to know if the post-hoc tests are appropriate.

4) The discussion of this study suffers from lack of a thorough review of previously published literature to place these findings in context. A few examples: First, the cited study on line 477 is from induced SGN loss in a single cat. More information is needed here, as there are numerous studies showing that aging results in SGN loss that contributes to elevated hearing thresholds (See Frisna et al., 2016, Age-related hearing loss: prevention of threshold declines, cell loss and apoptosis in spiral ganglion neurons). Second, there are no citations supporting the statement that thresholds could reflect HC loss and SV atrophy (line 479) , even though there are many studies examining the exact relationship between loss in these areas and age-related hearing loss (see Stebbins et al., 1979 or Lang et al., 2010).

Abstract:

Line 36-37: Please also state the number of young mice used.

Line 38- Please clarify what is meant by frequency characteristics of environmental sound.

Line 39- This is not accurate, as the environmental sound peaked at 1 kHz with a tapered decrease in intensity to around 10 kHz. It would be more accurate to just synapses, as loss of OHC’s and synapses were observed at all frequency regions (with OHC loss peaking at 11.3 kHz region) and synapse loss peaking at the 5.6 kHz region.

Introduction:

Line 58- Please use a different citation for the negative effect of tinnitus on quality of life, as the cited study just looks at incidence, not disease burden.

Line 71- Please use full CBA/CaJ strain label here.

Materials and Methods:

Line 87- Were these mice housed in the same environment for the duration of their lifespans?

Line 92-95- Were these recordings taken at multiple time points during the aged mouse lifespan?

Line 97-98- Were the young animals shipped for experiments as well?

Line 116- Were the ABR stimuli presented monaurally? Or binaurally? Was the reference electrode on the same side for each mouse?

Line 121-123- What was the starting dB value? How could sound levels be increased from <20 dB below threshold if the threshold of the animal was not determined yet?

Line 122- should be ‘=80 dB SPL’, as the measurements of wave I amplitude at latency were performed with data from 80 dB SPL stimuli.

Line 123-125- Were reviewers of the waveforms blinded to age group?

Line 163-167- Please clarify the difference in these fluorescent microscopes. Why were two different microscopes used?

Line 203-204- It is not clear how missing HC’s were determined. The authors state that the lack of Myo7a-positive cell bodies was used as the criterion for missing HC’s, but was there another metric used? As it is presented, it is impossible to quantify that something is missing based on the fact that Myo7a is missing, as there is no way to confirm that it was expressed in that region originally. Another approach to this would be to normalize the percent survival of HC’s in aged mice by normalizing it to HC’s in young mice.

Line 240- Where 100 axons from the same mice? Or 100 axons within the age group across multiple mice?

Line 249-251- How many samples were taken from each age group? Were these the same mice used for audiometric and histological analysis?

Line 261- please add the specific methods used to normalize data to the young group. Also, please specifically state which analyses used normalized data, as this is important information for interpreting the data presented.

Lines 258-260- Please include the specific predictor variables used for the 2-way ANOVAs. From context in the results, there are frequency and age, but they need to be stated.

Results

Figure 2- ABR thresholds in older mice at 5.66 kHz were above the 80 dB SPL level used to calculate P1 amplitude and P1 latency. Does this mean that data from a subset of the aged mice was reported in 2C and 2D? How did the authors account for mice with thresholds above the 80 dB stimulation used for suprathreshold ABR analysis?

Line 318- Please replace the word ‘huge’ with more accurate language, such as ‘greatest loss’.

Figure 4a- Although stated in the caption, please add a legend for the antibodies used for the immunohistochemical images.

Figure 4b- Do the authors have any hypothesis why synapses at 8 kHz were minimally affected? Also, please change the y-axis to a percentage, similar to the y-axes in figure 3b/c

Line 349-351- These data should be frames slightly differently to be more understandable. Although p=0.052 is not significant, it is a trend towards significance. A statement should be made towards this end, stating that although the effect size is large, the null hypothesis cannot be rejected. Furthermore, reporting of the 2-way ANOVA results would be helpful here, as it would determine if reporting the post-hoc comparisons between ages at the apical turn are appropriate.

Discussion

Line 483- Please add relevant citations showing the known relationship between OHC loss and elevated DPOAE thresholds.

Line 487- Please change this citation to one that directly examines the relationships between auditory nerve activity and ABR wave I metrics.

Line 489-495- In this section the authors link loss of SGNs to reduced ABR wave I amplitudes. Please provide citations to published literature examining this relationship and further citations studying the relationship between HC synapses and wave I amplitude.

Line 497- Are there published studies of orphan synapses in aging or is this the first to document the minimal change in orphan synapses with aging?

Line 517-519- The middle of the cochlea does not correspond to low frequencies.

Line 538-539- It is not accurate to say that the environmental noise was in the low to middle frequencies of the mouse audible range, as the mouse audible range is from 1-1000 Hz with the highest sensitivity at 16 kHz (Reynolds et al., 2010). It would be more accurate to say these frequencies are at the lower range.

569- This line states a 60% SGN survival, while line 478 states at 50% SGN survival. This information should be in the results as well.

Conclusion

Line 630-632- These results do not suggest that the severe SGN loss is due to mitochondrial damage and stress, just that aged mice exhibit oxidative stress. At minimum, correlational analysis between these metrics would need to be done for this statement to be included.

Reviewer #4: In this manuscript, the authors examined peripheral hearing in aging female CBA/CaJ mice as well as corresponding histopathological changes in the cochlea and spiral ganglion cells at both the light and electron microscopic levels. They also measured ambient sound levels in their vivarium and serum levels of a marker of oxidative stress. They examined two groups – a group of young females at 9-14 weeks and old females at 29-31 months. They found marked elevation of pure tone ABR thresholds and latencies, diminished P1 ABR amplitudes, and elevated DPOAE thresholds. They also found significant loss of outer hair cells with minimal inner hair cell loss. Although claims are made for a low-frequency bias of the loss, most of the data point to across-the-board losses. There was also evidence for spiral ganglion cell loss, cochlear hydrops and thrombosis and no obvious changes in myelination. Serum markers of oxidative stress were elevated in the aged group.

Most of the findings reported in this study have previously been reported, but as the authors point out, this is likely the most comprehensive accounting of aging in CBA/CaJ mice. Overall, there is utility to this study, but there are many weaknesses, outlined below. The manuscript would also benefit from a detailed review to correct the many grammatical errors and awkward phrasing.

Major:

1. The numbers of animals used for many aspects of this study are too few to draw reasonable conclusions. Many of the negative findings (e.g., the lack of differences in myelination) are grossly underpowered (n=3 per group) and even when differences are observed, the n’s are often too small to have a confident estimate of the difference between groups.

2. Along the same lines, there are instances were non-independent data are analyzed with ANOVA, which assumes independent samples. For example, in the hair cell counts, in some cases two “ears” come from one animal (and are therefore not independent) and other individual “ears” are only from one animal. Adjustment to the statistical plan should be made.

3. The absence of an intermediate age group is a major weakness which limits our understanding of when these changes occur and the degree to which the findings in the young group represent a developmental trajectory.

4. The argument that ambient noise in their vivarium (which does seem rather loud) is responsible for hearing loss is not compelling. The ambient noise is quite low frequency and does not align well to the distribution of hearing loss, which is mostly uniform in these mice.

5. Were the blood samples taken from the same mice that hearing was measured in? If so, it would be important to calculate correlations between hearing metrics and this marker of oxidative stress.

Minor:

1. There are many typos, grammatical errors and awkward word choices. I have outlined a few here. A thorough editing is needed.

2. There is a run-on sentence on lines 32-34

3. The age of the young mice should be listed in the abstract

4. Delete the comma after “area” in line 38

5. Line 46 – change “due to” to “potentially associated with”

6. Line 83 – change “keeping environment” to something like “laboratory animal environment” or “vivarium”

7. Space between no. and 21051 in line 100.

8. Line 104 – change “noise” to “ambient sound”

9. SV and SM abbreviations are not used consistently. Please define them once and use throughout.

10. “Huge” is not an appropriate term to describe data. This word is used 3x. Please find another word.

11. Line 249 – delete “blood”

12. Please state how blood was collected. How long was blood left to sit before supernatant was collected?

13. Fig 5 – please label SM and SV

14. In figs where aged data is normalized to young data (eg Fig 5), please indicate if the aged findings were normalized to the average young value.

15. The phrase “raising environment” is not right. Perhaps “vivarium” is better.

Reviewer #5: (No Response)

7. PLOS authors have the option to publish the peer review history of their article (what does this mean? ). If published, this will include your full peer review and any attached files.

**Do you want your identity to be public for this peer review?** For information about this choice, including consent withdrawal, please see our Privacy Policy .

Reviewer #1: **Yes: ** Toru Miwa

Reviewer #2: **Yes: ** Hao Xiong, MD & PhD

Reviewer #3: No

Reviewer #4: No

Reviewer #5: No

---

## [Author Response · Author response to Decision Letter 2]

2 Aug 2025

As described in the “Response_to_Reviewers” file.

---

## [Decision Letter · Decision Letter 2]

16 Aug 2025

PONE-D-25-21282R2Histopathological change of age-related hearing loss in female advance-aged CBA/CaJ micePLOS ONE

Dear Dr. Mizutari,

Thank you for submitting your manuscript to PLOS ONE. After careful consideration, we feel that it has merit but does not fully meet PLOS ONE’s publication criteria as it currently stands. Therefore, we invite you to submit a revised version of the manuscript that addresses the points raised during the review process.

We look forward to receiving your revised manuscript.

Kind regards,

Miriam A. Hickey, PhD

Academic Editor

PLOS ONE

Journal Requirements:

Additional Editor Comments:

Thank you for addressing comments from the Reviewers.

Please now address comments from Reviewer 3.

Please also address the following

1)

Please provide ANOVAs per mouse (N=5), as well as per ear for functional data.

(…Mice (n = 10 ears of 5 animals per group) were anesthetized intramuscularly using ketamine (75 mg/kg)…)

2)

Please amend phrase "three ears in both groups" (in Section Survival of cochlear HCs and synaptic ribbons; line 312 of manuscript with changes highlighted) to "three ears in two mice from both groups"

(Mice (n = 3 ears of 2 animals in each group) were euthanized... taken from line 148-149 of amended manuscript)

3)

For EM analyses, please provide statistics per mouse and per ear.

(The mice (n = 8 ears of 4 animals per group) were anesthetized...)

4)

Please clarify, in the "Statistical analyses"section, how the data from citation 11 was used for calculations of sample size.

Reviewers' comments:

Reviewer's Responses to Questions

**Comments to the Author**

1. If the authors have adequately addressed your comments raised in a previous round of review and you feel that this manuscript is now acceptable for publication, you may indicate that here to bypass the “Comments to the Author” section, enter your conflict of interest statement in the “Confidential to Editor” section, and submit your "Accept" recommendation.

Reviewer #1: All comments have been addressed

Reviewer #2: All comments have been addressed

Reviewer #3: (No Response)

Reviewer #4: (No Response)

2. Is the manuscript technically sound, and do the data support the conclusions?

Reviewer #1: Yes

Reviewer #2: Yes

Reviewer #3: Yes

Reviewer #4: No

3. Has the statistical analysis been performed appropriately and rigorously? 

Reviewer #1: Yes

Reviewer #2: Yes

Reviewer #3: Yes

Reviewer #4: No

4. Have the authors made all data underlying the findings in their manuscript fully available?

Reviewer #1: Yes

Reviewer #2: Yes

Reviewer #3: Yes

Reviewer #4: Yes

5. Is the manuscript presented in an intelligible fashion and written in standard English?

Reviewer #1: Yes

Reviewer #2: Yes

Reviewer #3: Yes

Reviewer #4: Yes

6. Review Comments to the Author

Reviewer #1: (No Response)

Reviewer #2: The authors have successfully addressed all concerns raised previously and the manuscript is suitable for publication in its current form.

Reviewer #3: I want to thank the authors for the time taken to respond to all of the comments. There was a lot of work done on the paper that has significantly improved it. However, I still have a few concerns with this manuscript.

1) Please clarify why 20 dB SPL was the starting point for determining thresholds. Did no mice have thresholds below 20 dB SPL? If so, then this is inconsistent with the data presented in Figure 2 B, where the average threshold from young mice at 16 kHz is less than 20 dB. And if the loudest. Please make sure that your methods section is accurate.

2) Thank you for adding a correlation analysis between IHC synapses and ABR P1 amplitude and latency. However, I have a few concerns about the analysis and presentation of this data. First, please remove any regression lines that are not significant, as they do not mean anything and can cause the illusion of significance. Second, there is an inconsistency between the results reported in the text and the figure. The text states that IHC synapse counts at both 16 kHz and 22 kHz were significantly correlated, but the 22 kHz data is missing from figure 4D. I understand that this was likely done to keep the figure simple, but as only two frequencies show significant associations, I feel that it is important. Finally, the authors show that IHC synapse count is associated with ABR P1 amplitude at 16 kHz and 22 kHz only, not at the other four analyzed frequency regions. This is in contrast to some publications, that show a significant association between synapse loss and P1 amplitude across all frequencies in aged CBA/CaJ mice (See Sergeyenko et al., 2013), which were not discussed in this context.

3) Why were correlation analysis not performed for other data sets? For example, were there any associations between orphan ribbons, IHC count, or OHC count?

4) In this paper, the authors found a unique difference in synapse counts between regions with minimal loss at 8 kHz. This is an interesting finding, but I have to respectfully disagree with the author’s hypothesis that this occurs due to it being the boundary area between 5.6 and the rest of the cochlea as to me it does not make logical sense. Yes, the ambient sound measured in this study was shifted towards lower frequencies, but the spectrum shows a relatively high level of energy between the 10^3 and 10^4, where 8 kHz is. Furthermore, as the sound presented in Feng et al., is white noise and theoretically should have equal energy spread across the entire spectrum, including the lower frequencies. In fact, Feng et al. found significant IHC synapse loss throughout the cochlea, from 10%-90%, even if it the loss was greatest at the 30-70% regions. That is to say, there is no ‘boundary’ in either the frequency spectrum recorded by the authors or in the citation used to justify this hypothesis.

Minor

- How many animals had thresholds at 85 dB SPL and how did this exclusion criteria affect the sample sizes for other audiometric measurements?

Reviewer #4: None of my major concerns could be addressed by the authors. The very low sample size is a major problem, and I do not believe that it is possible to reach a statistical power of 80% with n=3. In addition, the authors have continued to include non-independent samples in their ANOVA, have not correctly interpreted (in my opinion) the impact of environmental noise and are not able to draw firm conclusions from their use of separate animals from which they obtained blood samples and from which they tested hearing. Finally, the lack of an intermediate aged group is a major limitation on the impact of any aging-related study.

7. PLOS authors have the option to publish the peer review history of their article (what does this mean? ). If published, this will include your full peer review and any attached files.

**Do you want your identity to be public for this peer review?** For information about this choice, including consent withdrawal, please see our Privacy Policy .

Reviewer #1: **Yes: ** Toru Miwa

Reviewer #2: No

Reviewer #3: No

Reviewer #4: No

---

## [Author Response · Author response to Decision Letter 3]

1 Sep 2025

Point to Point Response to the Reviewers

Manuscript Number: PONE-D-25-21282R2

We would like to thank the editor and the reviewers for their comments. We have re-edited and improved our manuscript as suggested. In the following, we provide a point-to-point response to every comment raised. We have highlighted portions of the manuscript that have been updated in response to the reviewers’ comments.

Editor:

1. Please provide ANOVAs per mouse (N=5), as well as per ear for functional data.

(…Mice (n = 10 ears of 5 animals per group) were anesthetized intramuscularly using ketamine (75 mg/kg)…)

Response: We performed two-way ANOVAs and post hoc tests using per-mouse analysis, in which the measured values from both ears of each mouse were averaged. The overall results were consistent with the per-ear analysis, except for ABR P1 latency at several frequencies, where statistical significance was not observed due to reduced power from the smaller effective sample size. We have added this description to the Methods section (P.15, L.269–273) and the Results section (P.17, L.312–315), as well as in the Supporting Information (S1 Fig and S2 Table).

2. Please amend phrase "three ears in both groups" (in Section Survival of cochlear HCs and synaptic ribbons; line 312 of manuscript with changes highlighted) to "three ears in two mice from both groups"

(Mice (n = 3 ears of 2 animals in each group) were euthanized... taken from line 148-149 of amended manuscript)

Response: This sentence was corrected following the editor’s recommendation. (P.18, L.330)

3. For EM analyses, please provide statistics per mouse and per ear.

(The mice (n = 8 ears of 4 animals per group) were anesthetized...)

Response: We performed two-way ANOVAs and post hoc tests in the per-mouse analysis, following the same approach used for the cochlear function test (Editor’s Comment #1). The results showed trends comparable to the per-ear analysis, with statistical significance detected in the same metrics. This description has been added to the Methods section (P.15, L.269–273) and the Results section (P.22, L.404–405; P.27, L.480–482), as well as in the Supporting Information (S3 Fig and S4 Table).

4. Please clarify, in the "Statistical analyses"section, how the data from citation 11 was used for calculations of sample size.

Response: We apologize for the insufficient detail in our original submission. A more detailed description has now been added to the manuscript (P.16, L.280–285).

Reviewer #3:

I want to thank the authors for the time taken to respond to all of the comments. There was a lot of work done on the paper that has significantly improved it. However, I still have a few concerns with this manuscript.

Response: We sincerely thank the reviewer for the careful reading of our manuscript and the constructive comments. The thoughtful feedback has greatly improved the quality and clarity of the work, and we are truly grateful for the time and effort devoted to this review.

In response to the additional concerns raised, we have carefully addressed each point and revised the manuscript accordingly. We believe these further revisions have strengthened the paper, and we hope that the changes satisfactorily resolve the remaining issues.

1. Please clarify why 20 dB SPL was the starting point for determining thresholds. Did no mice have thresholds below 20 dB SPL? If so, then this is inconsistent with the data presented in Figure 2 B, where the average threshold from young mice at 16 kHz is less than 20 dB. And if the loudest. Please make sure that your methods section is accurate.

Response: We apologize for the ambiguity in our previous description. In young mice, particularly at 16 kHz, robust ABR waveforms are often detected even at 20 dB SPL. In such cases, the ABR Peak Analysis Software automatically determines thresholds below 20 dB SPL, and we adopted these automatically calculated values as the final thresholds. To clarify this procedure, we have revised the Methods section (P.7, L.131–133).

2. Thank you for adding a correlation analysis between IHC synapses and ABR P1 amplitude and latency. However, I have a few concerns about the analysis and presentation of this data. First, please remove any regression lines that are not significant, as they do not mean anything and can cause the illusion of significance. Second, there is an inconsistency between the results reported in the text and the figure. The text states that IHC synapse counts at both 16 kHz and 22 kHz were significantly correlated, but the 22 kHz data is missing from figure 4D. I understand that this was likely done to keep the figure simple, but as only two frequencies show significant associations, I feel that it is important. Finally, the authors show that IHC synapse count is associated with ABR P1 amplitude at 16 kHz and 22 kHz only, not at the other four analyzed frequency regions. This is in contrast to some publications, that show a significant association between synapse loss and P1 amplitude across all frequencies in aged CBA/CaJ mice (See Sergeyenko et al., 2013), which were not discussed in this context.

Response: We apologize for the inappropriate presentation of the data and fully agree with the reviewer’s comment. Figures 4D and 4E have been revised accordingly. We also thank the reviewer for highlighting the inconsistency with the previous report. In our study, simple linear regression analysis of ABR P1 amplitude and synapse count showed statistically significant correlations only at 16 and 22 kHz, while other frequencies demonstrated a trend toward proportional relationships (p = 0.09 at 8 kHz; p = 0.08 at 32 kHz). We therefore speculate that the absence of intermediate-aged cohorts and the limited sample size may have reduced the power to detect significant correlations. This point has been added to the Discussion section (P.31, L.566–P.32, L.570).

3. Why were correlation analysis not performed for other data sets? For example, were there any associations between orphan ribbons, IHC count, or OHC count?

Response: We thank the reviewer for the constructive suggestion and have conducted correlation analyses for additional metrics. Significant correlations were also detected between IHC/OHC counts and ABR P1 amplitude (revised Fig. 3). A review of the literature indicates that IHC-specific loss leads to a reduction in the compound action potential1, while OHC loss and functional deficits are associated with reduced ABR P1 amplitude2,3. Based on these findings, we have considered the combined cochlear histological damage across these components as a contributing factor to the low ABR P1 amplitude observed in advanced-aged mice. This description and the corresponding citations have been added to the Discussion section (P.32, L.578–585).

Reference #1: Qiu C, Salvi R, Ding D, Burkard R. Inner hair cell loss leads to enhanced response amplitudes in auditory cortex of unanesthetized chinchillas: evidence for increased system gain. Hear Res. 2000; 139(1-2): 153-171. https://doi.org/10.1016/s0378-5955(99)00171-9. PMID: 10601720.

Reference #2: Jung J, Joo SY, Min H, Roh JW, Kim KA, Ma JH, et al. MYH1 deficiency disrupts outer hair cell electromotility, resulting in hearing loss. Exp Mol Med. 2024; 56(11): 2423-2435. https://doi.org/10.1038/s12276-024-01338-4. PMID: 39482536.

Reference #3: Ingersoll MA, Lutze RD, Pushpan CK, Kelmann RG, Liu H, May MT, et al. Dabrafenib protects from cisplatin-induced hearing loss in a clinically relevant mouse model. JCI Insight. 2023; 8(24). https://doi.org/10.1172/jci.insight.171140. PMID: 37934596.

4. In this paper, the authors found a unique difference in synapse counts between regions with minimal loss at 8 kHz. This is an interesting finding, but I have to respectfully disagree with the author’s hypothesis that this occurs due to it being the boundary area between 5.6 and the rest of the cochlea as to me it does not make logical sense. Yes, the ambient sound measured in this study was shifted towards lower frequencies, but the spectrum shows a relatively high level of energy between the 10^3 and 10^4, where 8 kHz is. Furthermore, as the sound presented in Feng et al., is white noise and theoretically should have equal energy spread across the entire spectrum, including the lower frequencies. In fact, Feng et al. found significant IHC synapse loss throughout the cochlea, from 10%-90%, even if it the loss was greatest at the 30-70% regions. That is to say, there is no ‘boundary’ in either the frequency spectrum recorded by the authors or in the citation used to justify this hypothesis.

Response: We thank the reviewer for the insightful comment and agree that our speculation regarding 8 kHz as a potential boundary area where synapses are less vulnerable to acoustic stimuli was not sufficiently supported. Upon careful review of our results and the relevant literature, we found it challenging to establish a reasonable hypothesis confirming that 8 kHz synapses are least susceptible to impairment from previously reported pathologies. We therefore suspect that unknown pathophysiological mechanisms, potentially influenced by combined factors such as individual variability, strain, sex, or rearing conditions (aside from ambient sound), may underlie this finding. Accordingly, we have eliminated the “boundary” hypothesis from the manuscript and revised the relevant sentences in the Discussion section (P.36, L.657–P.37, L.665).

Minor

- How many animals had thresholds at 85 dB SPL and how did this exclusion criteria affect the sample sizes for other audiometric measurements?

Response: We apologize for the ambiguity in the manuscript. In the aged group, we observed ABR thresholds of 85 dB SPL in four mice (eight ears) at 5.6 kHz, one mouse (two ears) at 8.0 kHz, one mouse (one ear) at 11.33 kHz, and one mouse (two ears) at 22.65 kHz. However, the exclusion criteria were applied only to ABR P1 metrics and not to threshold analysis. Therefore, the results already presented were not affected. To clarify this point, we have revised the Methods section accordingly (P.8, L.137–139).

---

## [Decision Letter · Decision Letter 3]

5 Sep 2025

PONE-D-25-21282R3Histopathological change of age-related hearing loss in female advance-aged CBA/CaJ micePLOS ONE

Dear Dr. Mizutari,

Thank you for submitting your manuscript to PLOS ONE. After careful consideration, we feel that it has merit but does not fully meet PLOS ONE’s publication criteria as it currently stands. Therefore, we invite you to submit a revised version of the manuscript that addresses the points raised during the review process.

We look forward to receiving your revised manuscript.

Kind regards,

Miriam A. Hickey, PhD

Academic Editor

PLOS ONE

Journal Requirements:

Additional Editor Comments:

Editor comments have been addressed.

Reviewer #3: Please address concerns of Reviewer #3.

Reviewers' comments:

Reviewer's Responses to Questions

**Comments to the Author**

1. If the authors have adequately addressed your comments raised in a previous round of review and you feel that this manuscript is now acceptable for publication, you may indicate that here to bypass the “Comments to the Author” section, enter your conflict of interest statement in the “Confidential to Editor” section, and submit your "Accept" recommendation.

Reviewer #3: (No Response)

2. Is the manuscript technically sound, and do the data support the conclusions?

Reviewer #3: No

3. Has the statistical analysis been performed appropriately and rigorously? 

Reviewer #3: No

4. Have the authors made all data underlying the findings in their manuscript fully available?

Reviewer #3: Yes

5. Is the manuscript presented in an intelligible fashion and written in standard English?

Reviewer #3: Yes

6. Review Comments to the Author

Reviewer #3: All of my major comments have previous rounds have been addressed. However, there is a major issue in this submission. The authors state that they used the ABR Peak Analysis Software from Mass Eye and Ear Peabody Laboratories Engineering Core. They state that this software can automatically calculate, or extrapolate, thresholds lower than 20 dB SPL which was the lowest sound presentation level tested. This was strange to me, as I did not know of any software that could do this. I reached out to the team members of the Engineering Core who state that this piece of software cannot return a threshold lower than any of the tested levels. There is an option to auto-threshold, but if it detects that the threshold is lower than the lowest level tested, it warns the user and does not assign a threshold. Therefore, it seems impossible that the authors used this software to collect the audiometric thresholds reported in Figure 2 which are lower than 20 dB SPL. Was there a different software or recording method used to derive these thresholds other than those reported in the methods?

Although not a major component of the paper, this is a major concern for me as, unless there is software, analysis, or other recordings not reported in the methods, this is misinterpreted data at minimum.

I have a few other minor comments:

1. Please include non-significant statistics, especially if they are mentioned in the discussion. For example, when comparing the correlation between P1 amplitude and synapse count, only the statistics for the significant frequencies are reported. Then in the discussion, it's stated that other frequencies exhibit a trend towards a significant correlation, but the statistics for these correlations (including the p values) are not stated.

2. Please cite Sergeyenko et al., 2013 on line 567 so that readers understand what previous report is being referred to.

7. PLOS authors have the option to publish the peer review history of their article (what does this mean? ). If published, this will include your full peer review and any attached files.

**Do you want your identity to be public for this peer review?** For information about this choice, including consent withdrawal, please see our Privacy Policy .

Reviewer #3: No

---

## [Author Response · Author response to Decision Letter 4]

6 Sep 2025

Point to Point Response to the Reviewers

Manuscript Number: PONE-D-25-21282R3

We would like to thank the editor and the reviewers for their comments. We have re-edited and improved our manuscript as suggested. In the following, we provide a point-to-point response to every comment raised. We have highlighted portions of the manuscript that have been updated in response to the reviewers’ comments.

Reviewer #3:

All of my major comments have previous rounds have been addressed. However, there is a major issue in this submission. The authors state that they used the ABR Peak Analysis Software from Mass Eye and Ear Peabody Laboratories Engineering Core. They state that this software can automatically calculate, or extrapolate, thresholds lower than 20 dB SPL which was the lowest sound presentation level tested. This was strange to me, as I did not know of any software that could do this. I reached out to the team members of the Engineering Core who state that this piece of software cannot return a threshold lower than any of the tested levels. There is an option to auto-threshold, but if it detects that the threshold is lower than the lowest level tested, it warns the user and does not assign a threshold. Therefore, it seems impossible that the authors used this software to collect the audiometric thresholds reported in Figure 2 which are lower than 20 dB SPL. Was there a different software or recording method used to derive these thresholds other than those reported in the methods?

Although not a major component of the paper, this is a major concern for me as, unless there is software, analysis, or other recordings not reported in the methods, this is misinterpreted data at minimum.

Response: We thank the reviewer for this insightful comment and fully agree that an accurate description of the methods is essential to avoid misinterpretation of data. In our study, we analyzed the data using the ABR Peak Analysis Software downloaded from the Eaton-Peabody Laboratories Engineering Core, applied to recordings obtained with the Cochlear Function Test Suite in our facility. We recognize, however, that extrapolating thresholds without a detectable ABR waveform may underestimate the true threshold. To address this, we redefined ABR thresholds as 20 dB SPL only when clear waveforms were observed at a 20 dB SPL stimulus. Accordingly, we have reanalyzed the data and revised the Methods (P.7, L.128) and Results (P.16, L.297–P.17, L.300) sections, along with the relevant figures, table, and Supporting Information. Importantly, these adjustments did not affect our main conclusions.

I have a few other minor comments:

1. Please include non-significant statistics, especially if they are mentioned in the discussion. For example, when comparing the correlation between P1 amplitude and synapse count, only the statistics for the significant frequencies are reported. Then in the discussion, it's stated that other frequencies exhibit a trend towards a significant correlation, but the statistics for these correlations (including the p values) are not stated.

Response: We apologize for the insufficient data presentation that limited the connection between the Results and Discussion. We agree with the reviewer’s comment and have now added the p values across all frequencies for the correlation analysis between P1 amplitude and synapse count in the Results section (P.20, L.353–355).

2. Please cite Sergeyenko et al., 2013 on line 567 so that readers understand what previous report is being referred to.

Response: We thank the reviewer for the constructive comment. We have added the citation in the sentence as the reviewer’s suggestion. (P.32, L.568)

---

## [Decision Letter · Decision Letter 4]

15 Sep 2025

PONE-D-25-21282R4Histopathological change of age-related hearing loss in female advance-aged CBA/CaJ micePLOS ONE

Dear Dr. Mizutari,

Thank you for submitting your manuscript to PLOS ONE. After careful consideration, we feel that it has merit but does not fully meet PLOS ONE’s publication criteria as it currently stands. Therefore, we invite you to submit a revised version of the manuscript that addresses the points raised during the review process.

We look forward to receiving your revised manuscript.

Kind regards,

Miriam A. Hickey, PhD

Academic Editor

PLOS ONE

Journal Requirements:

**Additional Editor Comments:**

Specifically: Please include the 20 dB SPL lower threshold limit in the limitations section of the discussion.

Reviewers' comments:

Reviewer's Responses to Questions

**Comments to the Author**

1. If the authors have adequately addressed your comments raised in a previous round of review and you feel that this manuscript is now acceptable for publication, you may indicate that here to bypass the “Comments to the Author” section, enter your conflict of interest statement in the “Confidential to Editor” section, and submit your "Accept" recommendation.

Reviewer #3: (No Response)

2. Is the manuscript technically sound, and do the data support the conclusions?

Reviewer #3: Yes

3. Has the statistical analysis been performed appropriately and rigorously? 

Reviewer #3: Yes

4. Have the authors made all data underlying the findings in their manuscript fully available?

Reviewer #3: Yes

5. Is the manuscript presented in an intelligible fashion and written in standard English?

Reviewer #3: Yes

6. Review Comments to the Author

Reviewer #3: This is not contingent to acceptance, but I recommend that the authors include their 20 dB SPL lower threshold limit in the limitations section of the discussion.

7. PLOS authors have the option to publish the peer review history of their article (what does this mean? ). If published, this will include your full peer review and any attached files.

**Do you want your identity to be public for this peer review?** For information about this choice, including consent withdrawal, please see our Privacy Policy .

Reviewer #3: No

---

## [Author Response · Author response to Decision Letter 5]

16 Sep 2025

Point to Point Response to the Reviewers

Manuscript Number: PONE-D-25-21282R4

We would like to thank the editor and the reviewer for their comments. We have re-edited and improved our manuscript as suggested. In the following, we provide a point-to-point response to every comment raised. We have highlighted portions of the manuscript that have been updated in response to the reviewers’ comments.

Editor: Specifically: Please include the 20 dB SPL lower threshold limit in the limitations section of the discussion.

Reviewer #3: This is not contingent to acceptance, but I recommend that the authors include their 20 dB SPL lower threshold limit in the limitations section of the discussion.

Response: We thank the editor and reviewer for this valuable comment. In response, we have added the following sentence to the limitations section:

“A limitation of this study is that the lowest ABR threshold was set at 20 dB SPL, which may have restricted the evaluation of potential lower thresholds, particularly in the younger group.” (P.40, L.729-P.41, L.731)

---

## [Editor Report · Decision Letter 5]

22 Sep 2025

Histopathological change of age-related hearing loss in female advance-aged CBA/CaJ mice

PONE-D-25-21282R5

Dear Dr. Mizutari,

We’re pleased to inform you that your manuscript has been judged scientifically suitable for publication and will be formally accepted for publication once it meets all outstanding technical requirements.

Kind regards,

Miriam A. Hickey, PhD

Academic Editor

PLOS ONE

Additional Editor Comments (optional):

All comments have now been addressed.
---

## [Editor Report · Acceptance letter]

PONE-D-25-21282R5

PLOS ONE

Dear Dr. Mizutari,

I'm pleased to inform you that your manuscript has been deemed suitable for publication in PLOS ONE. Congratulations! Your manuscript is now being handed over to our production team.

Kind regards,

on behalf of

Dr. Miriam A. Hickey

Academic Editor

PLOS ONE